# A nanoscale MOF-based heterogeneous catalytic system for the polymerization of *N*-carboxyanhydrides enables direct routes toward both polypeptides and related hybrid materials

Ying Liu[1,3], Zhongwu Ren[1,3], Nannan Zhang[1], Xiaoxin Yang[1], Qihua Wu[2], Zehong Cheng[1], Hang Xing [1]✉ & Yugang Bai [1]✉

Synthetic polypeptides have emerged as versatile tools in both materials science and biomedical engineering due to their tunable properties and biodegradability. While the advancements of *N*-carboxyanhydride (NCA) ring-opening polymerization (ROP) techniques have aimed to expedite polymerization and reduce environment sensitivity, the broader implications of such methods remain underexplored, and the integration of ROP products with other materials remains a challenge. Here, we show an approach inspired by the success of many heterogeneous catalysts, using nanoscale metal-organic frameworks (MOFs) as co-catalysts for NCA-ROP accelerated also by peptide helices in proximity. This heterogeneous approach offers multiple advantages, including fast kinetics, low environment sensitivity, catalyst recyclability, and seamless integration with hybrid materials preparation. The catalytic system not only streamlines the preparation of polypeptides and polypeptide-coated MOF complexes (MOF@polypeptide hybrids) but also preserves and enhances their homogeneity, processibility, and overall functionalities inherited from the constituting MOFs and polypeptides.

Benefiting from their inherent biodegradability and tunable properties, synthetic polypeptides have become versatile platforms and tools in materials science[1–3] and biomedical engineering[4–8]. The first report of living ring-opening polymerization (ROP) of amino acid *N*-carboxyanhydrides (NCAs) was achieved using organometallic catalysts[9], which lays the very foundation of this field. Recent developments in NCA-ROP strategies[10–22] have turned to techniques for accelerated polymerizations, so that the preparations can become less time-consuming and moisture-insensitive. However, since the major focus in the polypeptide field has been on synthetic methodologies due to the inherent difficulty in such syntheses, various follow-up issues, including the generality, the cost of such preparations, the potential contamination or tagging of peptide product by residual catalysts, are less explored for now. In particular, the integration of ROP products with other materials is scarcely considered[23], despite its explicit importance in materials science. While there have been successful stories of preparing hybrid polypeptide-based materials through non-covalent self-assembly[24] or ligand exchange[25], in most cases, the

[1]State Key Laboratory of Chemo-/Bio-Sensing and Chemometrics, School of Chemistry and Chemical Engineering, Hunan University, 2 South Lushan Road, 410082 Changsha, Hunan, China. [2]Jordan Valley Innovation Center, Missouri State University, 524 North Boonville Avenue, Springfield, MO 65806, USA. [3]These authors contributed equally: Ying Liu, Zhongwu Ren. ✉e-mail: hangxing@hnu.edu.cn; baiyugang@hnu.edu.cn

integration of polypeptides with other components still asks for prior installation of highly active molecular connectors on both the polypeptides and the target material substrates[26–29]. Alternatively, the establishment of a viable surface-initiation system with additional efforts may do the job[26,30–34], but such systems barely benefit from modern NCA-ROP techniques and still likely suffer from the inconveniences such as moisture sensitivity and slow polymerization rate.

The success of the Ziegler-Natta catalyst suggests a potential solution for the above issues associated with NCA-ROP: metal-based heterogeneous catalysis[35]. In a typical heterogeneous catalytic process, monomers are absorbed into the insoluble metal catalysts and polymerized before eventually detaching as polymer chains. In this way, heterogeneous catalysis facilitates large-scale production by enabling easy isolation of products, as well as high turnover rates and the recyclability of the catalysts. Moreover, it can be speculated that if the final desorption step is not enabled, the catalyst-polymer complex naturally an inorganic@polymer hybrid that opens up alternative potential applications. Thus, we propose to establish a heterogeneous strategy for catalyzed NCA-ROP, which would allow a single method to be utilized for the preparation of both free polypeptides and polypeptide-based hybrid materials (Fig. 1). In this article, we show that nanoscale metal-organic frameworks (MOFs) can be utilized as heterogeneous co-catalysts in this system. As part of the heterogeneous catalytic system for NCA-ROP, the nanoscale MOFs, together with the in situ-formed polypeptide helix arrays, enable all the advantages that modern accelerated NCA-ROP techniques can offer in addition to their benefits of easy work-up and co-catalyst recyclability. More importantly, the catalytic system is versatile enough to allow direct preparation of both polypeptides and related MOF@polypeptide hybrid materials. The hybrid material prepared in this way shows distinct homogeneity, processibility, and performance compared to polypeptide or MOF alone or their simple mixture. Benefiting from the broad utility that MOFs present, this MOF-based heterogeneous strategy empowers the products generated from accelerated NCA-ROPs with tremendous potential as inseparable parts of diverse functional materials.

## Results
### Catalyst design
To design a strategy that leads to both efficient polypeptide synthesis and useful hybrid materials, two major considerations are needed: (1) a practical catalytic path must be established to allow an accelerated NCA-ROP; (2) the catalyst itself should have the potential of serving as a part of functional materials so that it can contribute to the utility of resulting hybrid materials. For the former, we consider the utilization of the reported autocatalytic effect of densely packed helical polypeptides from NCA-ROP[11,15,18], as a highly viable path. As the use of a heterogeneous catalyst will certainly lead to the packing of polymer chains on the active metal sites on the catalyst surface, it is likely that such an autocatalytic effect can be integrated into a two-step, heterogeneous catalytic system: a co-catalyst can just be responsible for the initiation part, while the chain propagation is aided by the in situ-formed array of polypeptide helices generated after chain initiation. The advantage of such a two-step design is evident. As the co-catalyst-mediated initiation (first step) of NCA-ROP is simply a nucleophilic attack toward an activated carbonyl group, candidates for such catalysis are many. This also means that the metal catalytic species are not at the propagating chain ends[36,37], otherwise, it would be difficult for the chains to easily detach from the catalyst. Instead of adopting a metal catalyst in the second step, the well-known, proximity-induced, self-catalyzed polymerization of NCA from packed polypeptide helices is utilized[11,14,18], thus assuring an accelerated chain propagation. Such a two-step catalytic approach will greatly simplify the design of the whole catalytic system, while retaining all the benefits that a heterogeneous catalytic system can offer.

For the latter, we turned to metal-organic framework (MOF) nanocrystals for a viable solution. MOFs can be viewed as close-packed metal complexes rich in coordinatively unsaturated metal sites on their surfaces (Fig. 2a). Considering the Lewis-acidic metal sites on MOFs[38,39] have the capability of activating carbonyl groups[40,41], we expect the combination of Lewis-acidic sites and a nucleophilic initiator, such as water, can initiate the two-step catalysis for NCA-ROP as mentioned above[42–45]. As the MOFs are co-catalysts rather than initiators, the resulting polypeptide chains are not linked to the MOF surfaces via strong covalent bonds. This characteristic feature makes it easy for them to detach from the heterogeneous co-catalyst, facilitating product isolation and co-catalyst recycling[46,47] when free polypeptide products are desired (Fig. 2b). Meanwhile, the high density of those coordinatively unsaturated metal sites on the surface is directly determined by the lattice structure of MOFs[48], which ensures the generation of densely packed polypeptide arrays capable of self-acceleration on MOF surface. Most importantly, MOFs are a family of well-recognized functional inorganic materials with versatile applications in separation, storage, electrocatalysis, delivery, and many other areas. The successful utilization of MOFs as co-catalysts will surely infuse the resulting catalyst@polypeptide complex, i.e., the hybrid material generated directly through this strategy, with diverse and attractive features for potential applications in various ways.

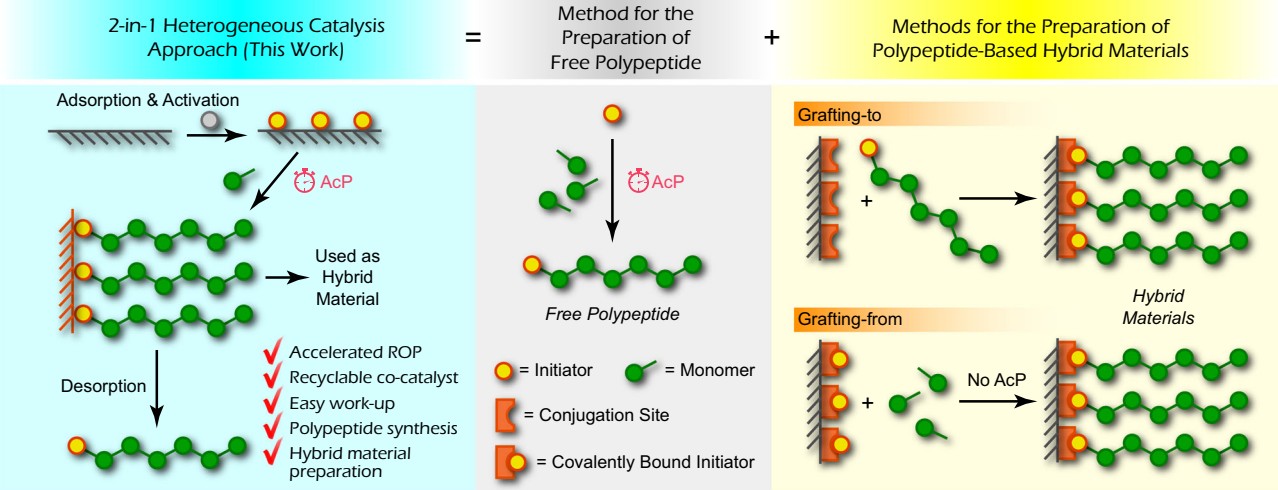

**Fig. 1 | Schematic illustration of the 2-in-1 design of the heterogeneous catalytic system presented in this work.** This unique system design allows both accelerated polymerization (AcP) of NCAs and one-step preparation of polypeptide-based hybrid materials.

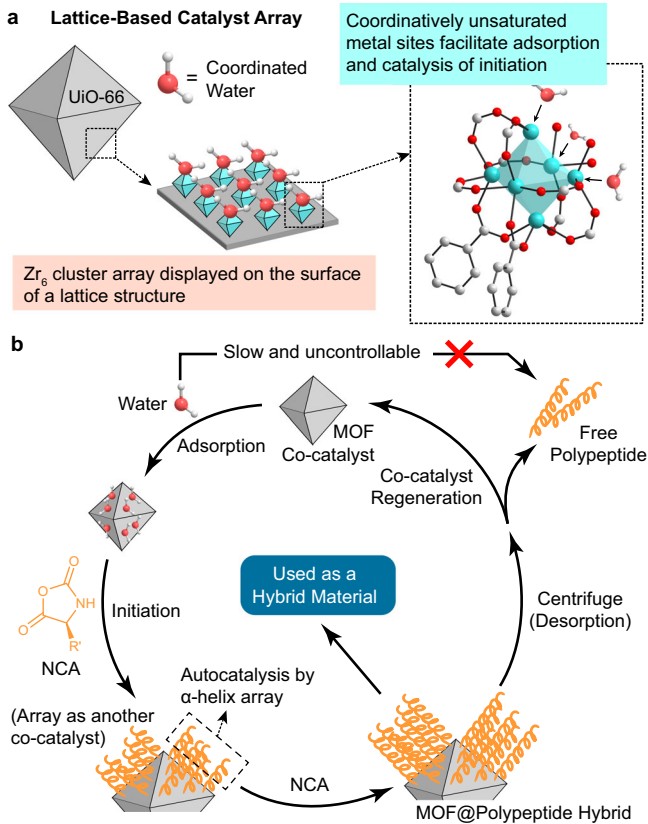

**Fig. 2 | Cartoon illustration of the initiator arrays that lead to proximity-induced accelerated polymerizations. a** Nanoscale UiO-66, as a representative MOF material, presents the surface-exposed active metal center ($Zr_6$) array. The structure and distribution of these catalytically active clusters are determined by the crystal lattice. **b** Proposed working cycle of MOF-aided accelerated preparation of free polypeptides and MOF@polypeptide hybrid materials.

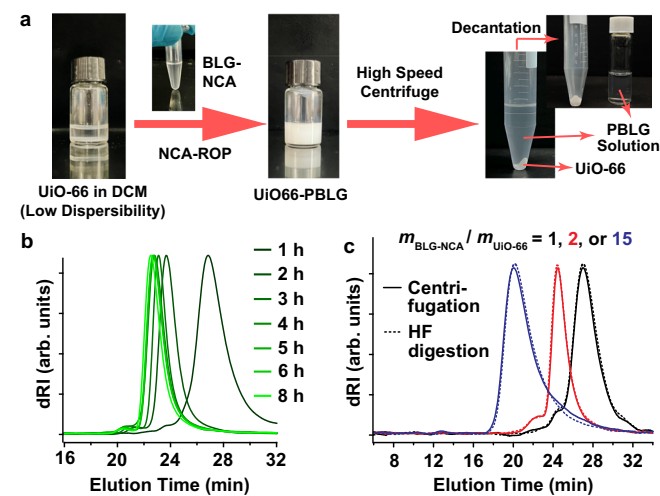

**Fig. 3 | Observations and characterization results in the study of UiO-66 NP-mediated NCA-ROP process. a** Photos taken during the course of polypeptide preparation using UiO-66 NP-mediated NCA-ROP. **b** GPC curves of PBLGs collected during the course of NCA-ROP with $m_{BLG\text{-}NCA}/m_{UiO\text{-}66} = 10$ and [BLG-NCA]$_0$ = 0.114 M. **c** GPC analyses showing that polypeptides obtained through centrifugation were identical to those obtained through HF digestion of MOF cores.

## UiO-66 nanoparticle mediates the ROP of BLG-NCA efficiently

We began to validate our design by preparing NPs of several classic MOFs, namely UiO-66 ($Zr^{4+}$), UiO-67 ($Zr^{4+}$), ZIF-8 ($Zn^{2+}$), NU-1000 ($Zr^{4+}$), HKUST-1 ($Cu^{2+}$), and MIL-101 ($Cr^{3+}$) (Supplementary Fig. 1), for a preliminary evaluation on their ability to modulate the ROP of γ-benzyl-L-glutamate (BLG) NCA in dichloromethane (DCM). After the resulting products were released, it was observed that the MOF particles with metal centers of higher Lewis acidity ($Zr^{4+}$, $Zn^{2+}$) could lead to NCA-ROP while the others ($Cr^{3+}$, $Cu^{2+}$) could not (Supplementary Fig. 2). Such a result supported the important role of Lewis acidity in heterogeneous NCA-ROP catalysis. Among those working co-catalysts, UiO-66 NPs showed the best control in poly(BLG) (PBLG) preparation as revealed by gel permeation chromatography (GPC). Thus, it was chosen as the model system for further studies. A series of UiO-66 NPs with varied diameters ranging from ca. 60 nm to 600 nm was thus prepared using acetate as modulator (60–80, 120–150, 200–250, and 550–600 nm for UiO-66-1/-2/-3/-4, respectively), so that more detailed studies were allowed (Supplementary Fig. 3).

Based on our design, the polymerization should have proceeded on the MOF surface as depicted by Fig. 1c, and this hypothetic mechanism was supported by several observations. First of all, the most characteristic features of heterogeneous catalysis, i.e., adsorption and desorption, could be observed for this MOF NP-based catalytic system. The absorption was revealed during polymerizations mediated by UiO-66, as evident changes in the dispersibility of these MOF NPs could be observed (Fig. 3a), suggesting the generation of polypeptide-on-UiO-66 hybrid complexes as NCAs were absorbed and polymerized on the NP surfaces. The molecular weight ($M_n$) of the

resulting polypeptide increased as time elapsed, consistent with the MOF NP's dispersibility change (Fig. 3b and Supplementary Fig. 4). The desorption was observed when the polypeptides were detached mechanochemically from the UiO-66 NPs via high-speed centrifugation (>13,000 rcf) to give naked MOF NPs (Supplementary Fig. 5): apparently, the formed polypeptides only weakly bonded to the MOF surface. The desorbed free polymers showed identical GPC chromatograms as those from HF etching (Fig. 3c), indicating that such a mechanochemical desorption process was thorough and complete. As one can easily see here, the centrifugation-based isolation of synthesized polypeptides was much safer and more convenient, distinguishing this co-catalytic system from surface-initiated polymerization systems in which free polypeptides, if ever wanted, must be hard-cleaved from the initiators. Such convenience is directly related to a catalytic design: the MOF is not directly initiating but catalyzing the ROP initiation step like other heterogeneous polymerization catalysts, therefore the polymer chains are only weakly bound, rather than connected to the MOF surface with strong covalent bonds.

## Verification of the two-step mechanism of UiO-66-involved catalysis

The hypothesized two-step catalysis, i.e., MOF-catalyzed initiation then propagation promoted by auto-catalysis of helical polypeptides formed (Fig. 2b), was also supported experimentally. First of all, the centrifugation supernatant after UiO-66-mediated NCA-ROP was analyzed by inductively coupled plasma mass spectrometry, which verified that no Zr species leaked during the polymerization and work-up stages. In fact, neither could simple Zr salts, such as $ZrOCl_2$ (used as the precursor for UiO-66), initiate the NCA polymerization (Supplementary Fig. 6). Therefore, it was clear that the MOF NPs played the catalytic role in our system. End-group analysis of the yielded PBLG was then conducted, using short peptides prepared at a low $m_{BLG\text{-}NCA}/m_{UiO\text{-}66}$ ratio and quenched at low monomer conversion. Matrix-assisted laser desorption/ionization time-of-flight mass spectroscopy (MALDI-TOF MS) revealed main series of peaks corresponded to polypeptides of different lengths with amine and carboxylic acid terminus, indicating that the chains were indeed initiated by water as designed (Fig. 4a). This initiation mechanism was also supported by the observation of ethyl

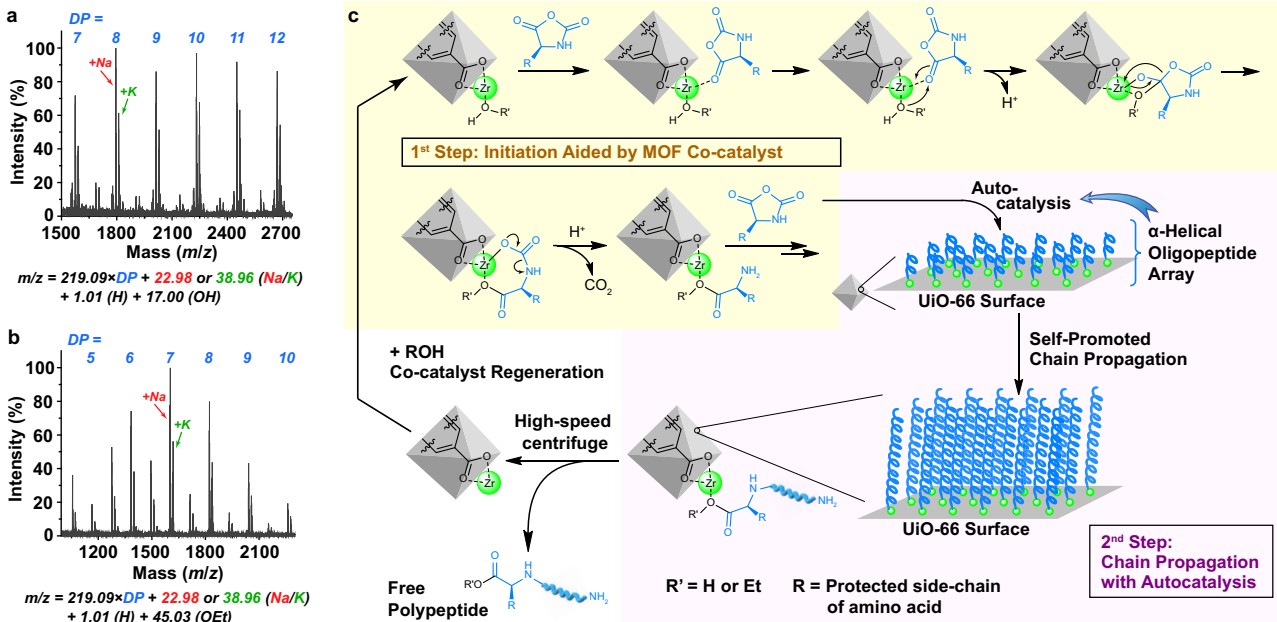

**Fig. 4 | Mechanistic studies on the MOF-catalyzed initiation process. a** MALDI-TOF MS spectrum of PBDLG with low molecular weight from water-initiated NCA-ROP on UiO-66 NPs. The polymerization was quenched shortly after initiation. **b** MALDI-TOF MS spectrum of PBLG from ethanol-initiated NCA-ROP on UiO-66 NPs, quenched shortly after initiation. **c** Detailed schematic illustration of the proposed mechanism of water-initiated NCA-ROP on UiO-66 NP. Enlarged spectra of panels **a** and **c** and detailed peak assignments are provided as Supplementary Figs. 7/8 in Supplementary Information.

ester-terminated oligomeric PBLG on MALDI-TOF spectrum when ethanol-washed UiO-66 NP was used in a similar manner (Fig. 4b).

It is known that although water alone could lead to the initiation of NCA-ROP, such initiation was far from efficient and controllable[15]. Thus, UiO-66 NP must have played a role in the initiation step. Indeed, it is long known that lithium and transition metal complexes with Lewis acidity are carbonyl activators, and can serve as catalysts for the ring-opening polymerizations[49,50] of cyclic esters[51–54], carbonates[55], and $O$-carboxyanhydrides[56,57] through a coordination-insertion mechanism. The UiO-66 NP is believed to be working in a similar way (Fig. 4c, yellow background part): the densely packed open Zr centers of UiO-66 NP activate the 5' carbonyl of NCAs, and the coordinated water molecules in close proximity can attack the carbonyl to give orthoesters which eventually eliminate to give the ring-opened products coordinated to the MOF NP surface through Zr-O dative bonds. Upon spontaneous decomposition of the resulting carbamic acid that releases $CO_2$, the 1st step, i.e., water initiation, is complete, and initial chain propagation continues via the normal amine mechanism to begin the 2nd step, giving the α-helical oligopeptide array capable of mediating subsequent autocatalyzed polymerization.

The presence of the adsorption-caused autocatalytic stage after MOF-aided initiation (Fig. 4c, pink background part) is revealed by the different consequences from the ROPs of monomers of different chirality. This initiation system showed no stereoselectivity for NCA monomers since the molecular weights and polymerization rate of the BDG-NCA were similar to those of BLG-NCA (Supplementary Fig. 9). However, a racemic mixture of BD/LG-NCA exhibited very different behavior in the polymerization, with a significantly reduced NCA conversion rate (68% conversion in 6 h, Supplementary Fig. 9). This is because that the MOF NP is only responsible for the catalysis of chain initiation, but the chain propagation is promoted by the auto-catalysis effect from the formation of densely packed helical polypeptides thereafter. Due to the inability to form helical structures by PBDLG, the self-acceleration in the propagation stage is nonexistent when racemic monomers are used[11]. As parallel evidence, the UiO-66 NP-mediated polymerization favors solvents with a low dielectric constant (chloroform and DCM, Supplementary Fig. 10) similar to the previous report[14], as solvent permittivity is a key factor affecting the self-acceleration behavior in NCA polymerization. With the desired acceleration in DCM, such MOF-mediated NCA-ROP could be conducted without an inert atmosphere or thoroughly dried solvent. Benchtop DCM without drying could be used without affecting the polymerization kinetics and the dispersity of the resulting polypeptides, although the molecular weights could turn slightly lower compared to the polymers prepared at the same $m_{BLG-NCA}/m_{UiO-66}$ ratio in anhydrous DCM (Supplementary Figs. 11 and S12). This was expected according to the proposed mechanism, as free water molecules in the solvent could coordinate to Zr centers by taking the empty sites or by replacing the acetate ligands[42,58], so that more initiation sites were generated, leading to a lower monomer to initiator ratio and lower $M_n$.

### Kinetic features of the MOF-mediated NCA-ROP

To further reveal the nature of this accelerated ROP stage, we studied the polymerization kinetics using infrared (IR) spectroscopy. As shown in Fig. 5a, two-stage, sigmoidal kinetic curves were observed for all $m_{BLG-NCA}/m_{UiO-66}$ ratios, and such two-stage kinetic feature is characteristic of autocatalytic NCA-ROP reactions as reported by Cheng et al. [11,15]. At a low NCA/MOF ratio ($m_{BLG-NCA}/m_{UiO66} = 2.5$), the polymerization was significantly faster and could complete in 2 h. Monitoring the consumption of BLG-NCA monomer at different $m_{BLG-NCA}/m_{UiO-66}$ ratios with [1]H NMR resulted in similar trends (Supplementary Fig. 13). Such reactions were significantly faster than trimethylsilyl amine-initiated BLG-NCA's ROP in DCM (Supplementary Table 1), indicating a cooperative, self-promoting behavior from neighboring helical polypeptides similar to the phenomenon reported in the literature[11,15]. Plotting $\ln([NCA]_0/[NCA]_t)$ against time in these polymerizations resulted in Fig. 5b, from which one can clearly see that the polymerizations followed a pseudo-first-order kinetics in the accelerated stage. The calculated $k_{p, acc}$ was larger for reactions done with more MOF NPs, because more NPs meant more chains growing at the same time. The number of growing chains, equivalent to the number of effective initiation sites (EIS) on the MOF NPs, can be quantified based

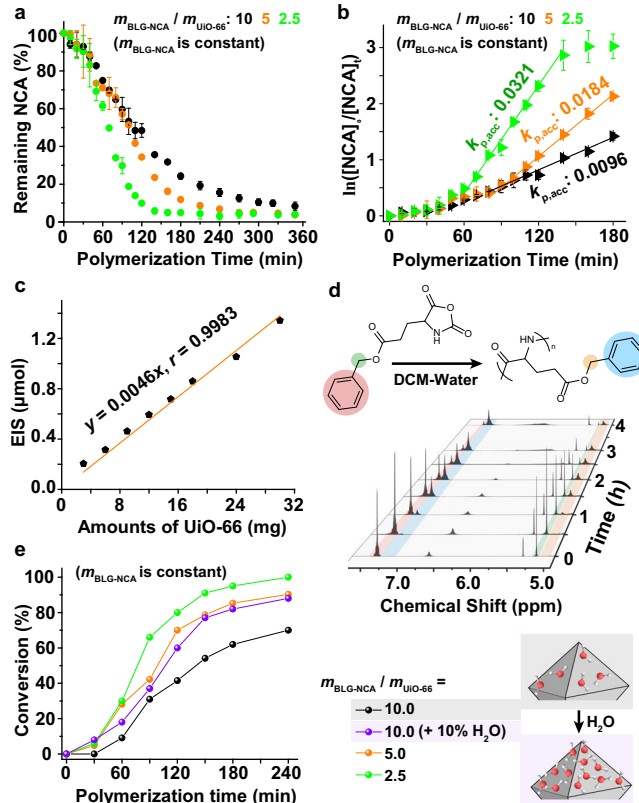

**Fig. 5 | Kinetic studies of MOF-mediated NCA-ROP under different conditions.** **a** Kinetic analyses of the UiO-66-1 NP-mediated NCA-ROP using IR spectroscopy. Polymerizations were performed with different $m_{BLG-NCA}/m_{UiO-66}$ ratios and fixed $[BLG-NCA]_0 = 0.114$ M in anhydrous DCM. **b** The log-based plot of monomer conversion versus time shows the differences in reaction rates for NCA-ROPs in the acceleration stage when the amount of UiO-66 changed. The same dataset as in panel **a** was used. **c** The fitting plot of total EIS on the surfaces versus the amount of UiO-66-2 NPs used, in the range of $m_{BLG-NCA}/m_{UiO-66} = 1–10$. **d** In situ $^1$H NMR monitoring of the BLG-NCA polymerization ($m_{BLG-NCA}/m_{UiO-66} = 10$) in $CD_2Cl_2$ containing 10% water. **e** Conversion-versus-time plot of the ROP of BLG-NCA under different conditions, calculated from peak integrations using the spectra obtained during the $^1$H NMR-based kinetic study. Error bars represent ± SD.

on the characterized DP of the resulting polypeptide product:

$$[EIS] = [NCA]_0 \times \text{conversion}/(\text{DP of the resulting polypeptide}) \quad (1)$$

When $m_{BLG-NCA}/m_{UiO-66}$ was set between 1 and 10, a linear relationship was clearly observed by plotting the used amounts of UiO-66 NPs and the total EIS on the particle surface (Fig. 5c). This indicates that in this $m_{BLG-NCA}/m_{UiO-66}$ range, the total EIS for NCA-ROP was proportional to the MOF NP used. Thus, the rate of each chain propagation reaction ($PBLG_n + BLG-NCA \rightarrow PBLG_{n+1} + CO_2$) could be described as:

$$-d[NCA]/dt = k_{ROP}[NCA][PBLG] = k_{ROP}[NCA][EIS], \quad (2)$$

and

$$k_{p,acc} = k_{ROP}[PBLG] = k_{ROP}[EIS] \quad (3)$$

The bimolecular reaction rate of the ring-opening process in the acceleration stage, i.e., $k_{ROP}$, could be calculated. $k_{ROP}$ was observed as almost a constant regardless of the amount of MOF used for initiation due to unchanged $d_{EIS}$. As exemplified in Fig. 5b, c, with quadrupled amounts of UiO-66 NPs used in the polymerization (black curve

to green curve), both the $k_{p,acc}$ and [EIS] increased, by 230% (0.0096 min$^{-1}$ to 0.0321 min$^{-1}$) and 190% (0.205 mM to 0.594 mM in 1 mL of DCM), respectively. Thus $k_{ROP}$, calculated as $k_{ROP} = k_{p,acc}/[EIS]$, showed a negligible change of 14%, which was in the typical error range of such kinetic measurements. Apparently, the overall rate accelerated by the helices, represented by $k_{p,acc}$, was not only determined by the helices' density[11] that was related to MOF's lattice and NCA structure, but also multiplied by the number of initiation sites [EIS] as Equation (3) had indicated. [EIS] was also proportional to $m_{UiO-66}$ used in a certain range of $m_{BLG-NCA}/m_{UiO-66}$ ratios. However, it should be noted that for UiO-66-NPs of other sizes, the EIS linear range may be different.

Straightforwardly, the total number of EIS in this system is directly related to the water present in the system, because water is the actual initiator on the UiO-66 surface according to the proposed mechanism. Thus, in addition to increasing the amount of UiO-66 NP (which carries residual $H_2O$ bound on the surface) used in the polymerization, the introduction of extra water into the system should increase total EIS. It was mentioned earlier that when the wet solvent was used for NCA-ROP mediated by those MOF NPs, the resulting polypeptides were shorter because more chains were generated. This observation was also true when bulk water was introduced: even a biphasic DCM/water mixture could be used as the medium for NCA-ROP, in which process BLG-NCA polymerized smoothly with the presence of UiO-66 NPs, yielding well-defined PBLG in 4 h ($Đ < 1.2$, Fig. 5d and Supplementary Fig. 14a). Not surprisingly, sharply decreased molecular weight was also observed for the resulting polypeptide when extra water was introduced (Supplementary Fig. 14b) because water led to increased [EIS] and more chains produced. Since the presence of phase-separated water would saturate all the coordination sites on UiO-66 NPs' surfaces, $M_n$'s of the products were similar for reactions with 10% or 20% water added. Importantly, as displayed in Fig. 5e, 10% water in the biphasic medium made polymerization proceed faster because higher EIS density ($d_{EIS}$) resulted from more coordinated water on MOF could generate polypeptide arrays of higher density[11] and higher [EIS], both leading to higher $k_{p, acc}$ for the polymerization. Also, it should be noted that water-initiated controllable NCA-ROP was limited to the UiO-66 NP-catalyzed process only: an "NCA+Water without MOF" system did not work. As the uncatalyzed NCA hydrolysis was much slower compared to the MOF-catalyzed initiation reaction and the autocatalyzed chain propagation reaction, good dispersity for the product polypeptides was still easily achievable in the presence of water when UiO-66 was used.

## Wide applicability of the heterogeneous polymerization strategy

All these observations shown above indicated that this heterogeneous catalytic system design was successful for efficient polypeptide preparation. To get further insights into the applicability of this robust system, the UiO-66 NPs with varied diameters (UiO-66-1/-2/-3/-4) were evaluated in different polymerization conditions. Using UiO-66-2 as the representative example, by increasing the feeding mass ratio ($m_{BLG-NCA}/m_{UiO-66} = 1–20$, $[BLG-NCA]_0 = 0.114$ M) in the ROP, all the synthesized PBLGs displayed unimodal peaks on GPC with decreasing $M_n$ and narrow dispersity (≤1.2) (Fig. 6a, b). The obtained $M_n$ of PBLG increased almost linearly when the $m_{BLG-NCA}/m_{UiO-66}$ ratio was increased from 1 to 10, but the slope slightly decreased for higher $m_{BLG-NCA}/m_{UiO-66}$ ratios. The results were similar for UiO-66 NPs of other sizes, but $M_n$ of synthesized PBLG slightly increased with larger UiO-66 NPs under the same condition (Supplementary Fig. 15), which was attributed to the decreased accessible surface area and number of initiation sites for larger NPs. Importantly, those NCA polymerizations mediated by all MOF NPs were highly predictable and reproducible (Supplementary Fig. 16), and the molecular weight of the PBLG product was solely dependent on the $m_{BLG-NCA}/m_{UiO-66}$ ratio regardless of monomer concentration used (Supplementary Fig. 17). As the

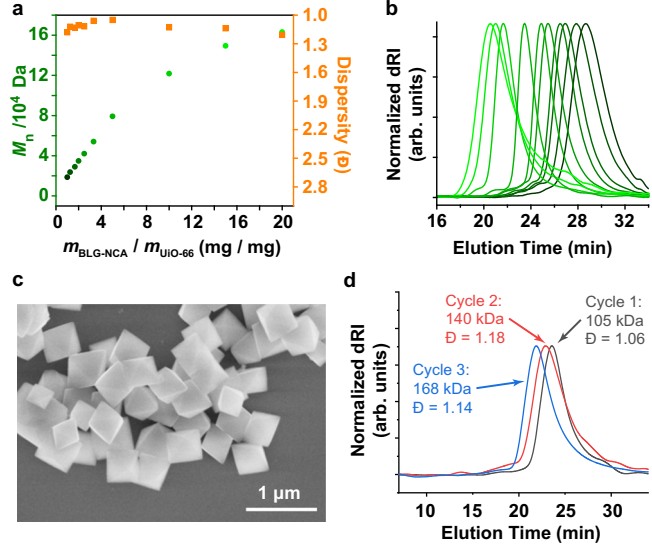

**Fig. 6 | Evaluation of polymerization controllability, catalyst batch variation, and catalyst recyclability. a** Molecular weights and dispersity control for the obtained PBLG at different polymerization conditions mediated by UiO-66-2. For all $m_{BLG-NCA}/m_{UiO-66}$ ratios, a fixed $[BLG-NCA]_0$ (0.114 M) in a fixed volume of anhydrous DCM was used. **b** GPC-dRI curves of the PBLGs are shown in panel **a**, with matching colors denoting the same dataset. **c** SEM visualization of UiO-66 NPs recovered after being used as a catalyst for NCA-ROP. **d** GPC-dRI curves, molecular weights, and dispersity information of PBLGs obtained using pristine and recycled MOF co-catalyst.

## Table 1 | NCA polymerizations initiated by nanoscale UiO-66[a]

| NCA | M/I Ratio[a] | $M_n$ (kDa) | Dispersity | DP | $d_{EIS}$[b] (µmol/mg) | Isolated Yield[c] (%) |
|---|---|---|---|---|---|---|
| BLG | 30/30 | 21.0 | 1.10 | 95 | 0.040 | 56 |
| BLG | 30/24 | 26.7 | 1.09 | 122 | 0.039 | 64 |
| BLG | 30/18 | 33.8 | 1.10 | 154 | 0.041 | 68 |
| BLG | 30/15 | 39.0 | 1.07 | 178 | 0.042 | 72 |
| BLG | 30/12 | 50.2 | 1.09 | 229 | 0.041 | 80 |
| BLG | 30/9 | 62.8 | 1.02 | 286 | 0.044 | 80 |
| BLG | 30/6 | 82.7 | 1.10 | 378 | 0.050 | 84 |
| BLG | 30/3 | 131.7 | 1.08 | 601 | 0.063 | 92 |
| BDG | 30/9 | 72.0 | 1.03 | 328 | 0.039 | 85 |
| BDG | 30/3 | 143.0 | 1.07 | 652 | 0.058 | 91 |
| BLA | 30/30 | 4.4 | 1.08 | 21 | 0.190 | 78 |
| BLA | 30/15 | 6.7 | 1.05 | 33 | 0.242 | 81 |
| BLA | 30/9 | 12.1 | 1.11 | 59 | 0.226 | 83 |
| ZLL | 30/9 | 56.1 | 1.08 | 214 | 0.051 | 88 |
| ZLL | 30/3 | 120.2 | 1.07 | 459 | 0.071 | 93 |
| BDLG[d] | 30/3 | 67.8 | 1.19 | 310 | 0.110 | 84 |
| BLA-r-BLG | 30/9 | 19.6 | 1.17 | 48 + 45 | 0.139 | 87 |
| BLA-b-BLG[e] | 30/9 | 13.5 | 1.07 | 33 + 31 | 0.203 | 85 |

[a]UiO-66-1 was used. Conditions: MOF: 3–30 mg, NCA: 30 mg, solvent: anhydrous DCM, r. t., 12 h to ensure full conversion of non-racemic monomers by FT-IR. Absolute $M_n$ and Đ were determined by GPC-MALLS.

[b] Density of effective initiation site on MOF surface, $d_{EIS} = n_{EIS}/m_{UiO-66}$. $n_{EIS} = n_{NCA} \times$ conversion/ (DP of the resulting polypeptide).

[c]Calculated by the collected PBLG after precipitation and washing in ether.

[d] Racemic mixture of BLG-NCA and BDG-NCA (1:1), 90% conversion by FT-IR after 12 h.

[e]Block copolymerization by sequential addition of BLA-NCA monomer (15 mg) and BLG-NCA monomer (15 mg), 6 h reaction time for each block. EIS was calculated based on BLA monomer.

---

facileness and reproducibility of MOF preparation are well-known, batch variations for MOF NPs are small enough to allow all the heterogeneously catalyzed polymerizations to be performed with satisfactory consistency and controllability: the MOF catalysts synthesized in different batches can maintain almost the same DP of the synthesized PBLG from BLG-NCA (Supplementary Fig. 18). The system can also be applied to other commonly used NCA monomers, such as $N^\varepsilon$-carboxybenzyl-L-lysine (ZLL) NCA, β-benzyl-L-aspartate (BLA) NCA, and γ-benzyl-D-glutamate (BDG) NCA. Characterization results for polypeptides were obtained using UiO-66-1 and NCAs of different ratios listed in Table 1. All the polymerizations produced well-defined polypeptides with narrow dispersity. In addition, when chain-end functionalizations are desired, amine-reactive small molecules could be added to modify the polypeptide N-terminus just like other NCA-ROP strategies (Supplementary Fig. 19).

### Recyclability of the MOF co-catalyst

As mentioned above, in all polymerizations, the resulting polypeptides were eventually isolated by centrifugation, leaving naked MOF nanoparticles. SEM characterization revealed that the morphology of those UiO-66 nanoparticles after serving as a co-catalyst for NCA-ROP remained mostly unchanged (Fig. 6c and Supplementary Fig. 5). Thus, it is intuitive for us to investigate if the MOF nanoparticles can be used repetitively for NCA-ROP, so that catalyst recyclability, as one of the most important features of heterogeneous catalysis, could be enabled. After being used in the polymerization, the UiO-66 NPs were regenerated by placing them in boiling water for 20 min, soaking in acetone for 8 h, then they were centrifuged and dried before used in another round of polymerization. The recovery yield of MOF was approximately 87% on average. Excitingly, GPC characterization indicated that the recycled MOF co-catalyst could also give well-defined polypeptides with narrow dispersity and controllable molecular weights based on $m_{BLG-NCA}/m_{UiO-66}$ ratios (Fig. 6d and Supplementary Fig. 20). It should be noted that the molecular weight slightly increased after repetitive

use, presumably due to the deactivation of some surface catalytic centers that led to decreased amount of adsorbed water and higher M/I ratios. Although such a phenomenon is almost inevitable for most heterogeneous catalysts, it can be easily countered in common synthetic practices by simply decreasing the $m_{BLG-NCA}/m_{UiO-66}$ ratio. Thus, considering the easy separation of co-catalysts from the polymers and straightforward regeneration procedures, this heterogeneous catalytic design has provided a viable path toward sustainable production of polypeptides, and the recyclability further enhanced its economic effectiveness (Supplementary Tab. 2 in Supplementary Information).

### A direct route to MOF@polypeptide hybrid material preparation

In addition to serving as a new strategy for heterogeneous catalytic system design and for polypeptide synthesis, we next aimed to demonstrate that the MOF@polypeptide complexes formed in situ during the NCA polymerization can be directly used as functional hybrid porous materials. It is generally acknowledged that surface functionalization of MOF particles is a feasible solution to improve the processibility of MOFs for practical applications[59–61]. Thus, we planned to use the UiO-66 NP-mediated in situ NCA polymerization as a scalable and straightforward route to directly prepare biocompatible and uniform MOF-polymer mixed matrix membrane (MMM) material[62] without the need for chain detachment or further post-functionalization. By controlling the $m_{BLG-NCA}/m_{UiO-66}$ ratio, we obtained uniform MOF@Polypeptide membranes with different MOF contents (10–37.5 wt.%) directly from the yielded UiO-66@PBLG hybrids. Scanning electron microscopy (SEM) images and energy-dispersive X-ray spectroscopy (EDX) element mapping revealed that the UiO-66 NPs were

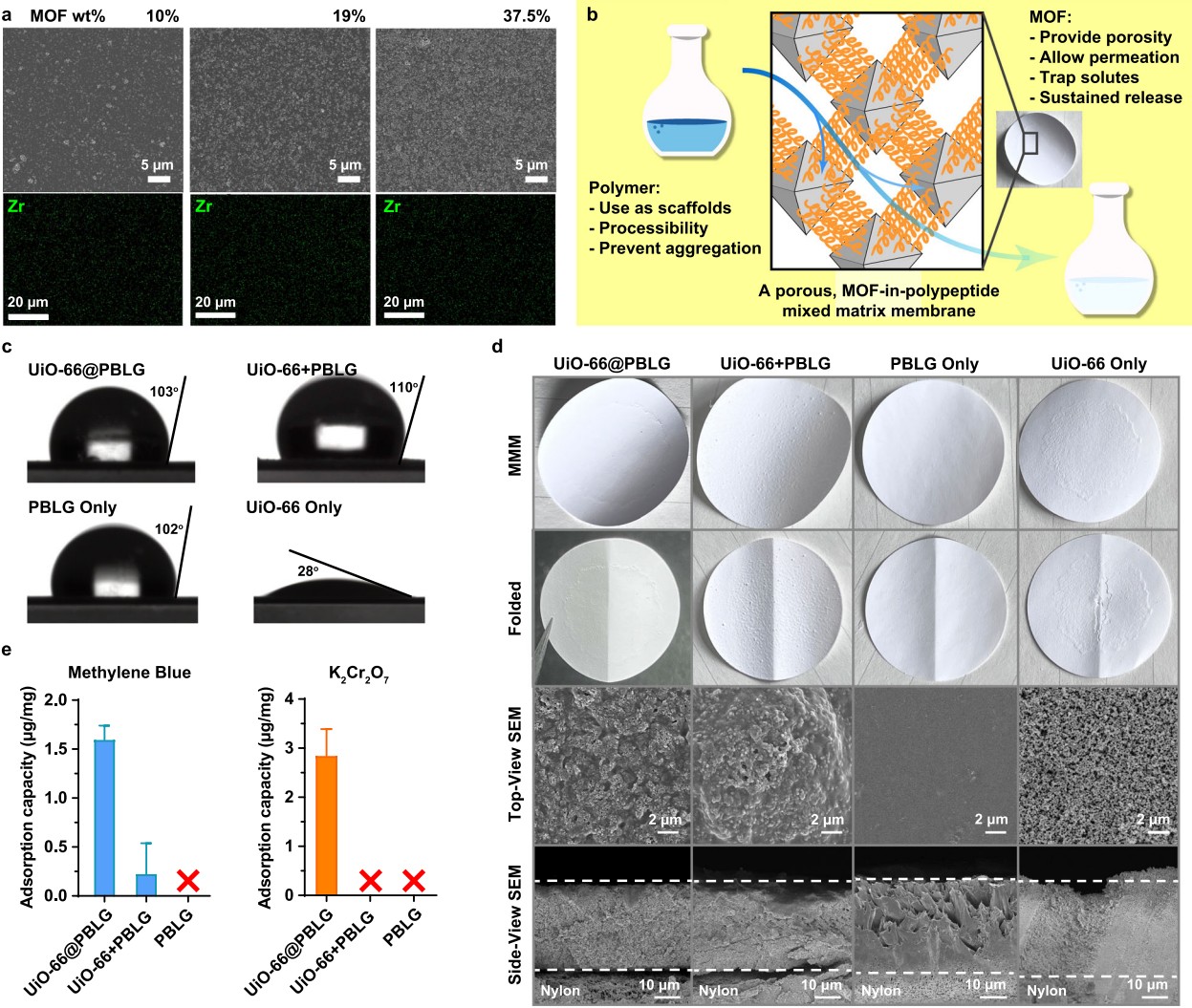

**Fig. 7 | Preparation, characterization, and evaluation of UiO-66@PBLG MMMs. a** SEM and EDX mapping images of UiO-66@PBLG, which were directly from the NCA-ROPs mediated by UiO-66 NPs. $[\text{BLG-NCA}]_0 = 0.114$ M, $m_{\text{BLG-NCA}} = 30$ mg, from left to right: $m_{\text{BLG-NCA}}/m_{\text{UiO-66}} = 10, 5, 2$, respectively. Numbers on the corners of the images denote the measured wt% of MOF in the films. **b** Cartoon illustration on the MMM based on UiO-66@PBLG. **c** Water contact angles of the prepared membranes. UiO-66@PBLG MMM was prepared directly from the ROP of BLG-NCA mediated by UiO-66 NPs ($m_{\text{BLG-NCA}}/m_{\text{UiO-66}} = 2$) containing 38.5 wt% UiO-66-1. UiO-66 + PBLG membranes were prepared by physically mixing 38.5 wt% UiO-66-1 NPs with pure PBLG of the same $M_n$. **d** Photographs of the prepared MMMs, folded MMMs, and top-/side-view SEM images of the MMMs. **e** Adsorption capacity comparisons of MMMs made from UiO-66@PBLG, UiO-66 + PBLG, and PBLG. Stable MMM could not be prepared using UiO-66 only thus it was not tested. × implies solution impermeability in the bar graphs. Error bars represent ± SD.

well dispersed in the material with high uniformity (Fig. 7a and Supplementary Figs. 21 and S24). Given the fact that other types of inorganic NPs can initiate NCA-ROP, this strategy has the potential to serve as a general route to prepare diverse polymeric-inorganic hybrid materials[31].

To highlight the advantages of MOF-mediated in situ NCA polymerization, we fabricated the MMMs using UiO-66@PBLG and demonstrated their potential as filter membranes (Fig. 7b). As previously stated, UiO-66@PBLG was prepared using in situ MOF-catalyzed NCA-ROP without co-catalyst (MOF) removal. To prepare the MMMs, UiO-66@PBLG in DCM right after polymerization was deposited onto a Nylon membrane substrate. After the removal of the organic solvent, UiO-66@PBLG particles became tightly bound to the Nylon substrate because of strong polar interactions and hydrogen bonding between the amide moieties on Nylon and PBLG chains. In the formed MMMs, the UiO-66 particles were automatically well dispersed inside the polymer matrix, owing to their initial coating with PBLG. As controls, we prepared membranes using PBLG only, UiO-66 NP only, and a simple mixture of PBLG and UiO-66 NP (UiO-66 + PBLG, from

purified, MOF-free PBLG and pristine UiO-66 NP). We have measured the water contact angles (WCAs) of all these materials. PBLG, UiO-66@PBLG, and UiO-66 + PBLG all showed similar WCAs while UiO-66 itself showed much higher hydrophilicity (Fig. 7c). However, photographs of the prepared membranes showed distinct characteristics (Fig. 7d and Supplementary Figs. 24 and S27). The UiO-66@PBLG and PBLG-only membranes appeared smooth appearance even after folding, while the UiO-66-only membrane had rough surfaces with poor inter-particle adhesion and adhesion to the Nylon substrate. A simple folding test clearly showed the vulnerability of such loosely packed structures: the UiO-66 layer near the folding line fell off easily. Such fragility suggested that the UiO-66-only membrane could not be used in any real property test. In contrast, UiO-66@PBLG and PBLG-only membranes were intact after folding. Regarding the UiO-66 + PBLG membrane, while no discernible fractures were observed in the folding, noticeable phase separation and nanoparticle aggregation were evident on the membrane, potentially impeding the membrane's overall performance. SEM images of the MMMs revealed that UiO-66 particles were homogeneously dispersed in the UiO-66@PBLG matrix

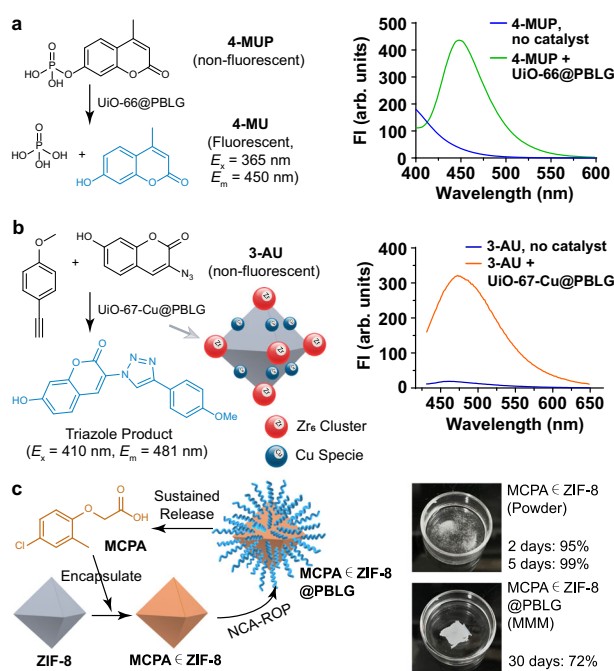

**Fig. 8 | Explorations on the potential applications of MOF@polypeptide MMM materials. a** Scheme and data showing the fluorogenic phosphate hydrolysis mediated by UiO-66@PBLG. **b** Scheme and data showing the fluorogenic CuAAC reaction mediated by UiO-67-Cu@PBLG. **c** Scheme and data showing that ZIF-8@PBLG MMM loaded with MCPA achieved long-lasting sustained release of the molecule of interest.

with uniform cross-sectional morphology, while the UiO-66 + PBLG simple mixture showed agglomerations and fractures in the cross-section (Fig. 7d). This is because the in situ NCA polymerization resulted in uniform PBLG coverage on each MOF particle, and the polymer chains were naturally aligned to allow stronger interactions with the inorganic nanomaterials as they were anchored on the inorganic surface since preparation. Presumably, these will minimize the potential of phase separation during MMM preparation. These results explicitly demonstrated the unique features of MOF-mediated in situ NCA polymerization in the preparation of functional MMMs.

Subsequently, we assessed the adsorption filter performance of the prepared MOF@Polypeptide membranes using solutions of methylene blue, an organic dye, and $K_2Cr_2O_7$, a metallate salt (Fig. 7e). Due to the significant fractures observed in the prepared pristine UiO-66 MMM, we evaluated the performance of UiO-66@PBLG, UiO-66 + PBLG simple mixture, and PBLG-only membranes. As a well-known feature of MOF materials, their intrinsic porosity allows for the absorption of small molecules and ions by trapping them within their internal voids. By subjecting the dye and $K_2Cr_2O_7$ solutions to vacuum filtration using MMMs as the filter, we observed clear and distinct absorption for UiO-66@PBLG membranes. In contrast, the UiO-66 + PBLG membrane exhibited only approximately 1/8 of the absorption effectiveness towards methylene blue and failed in filtering the $K_2Cr_2O_7$ solution as it could not pass through the membrane. The PBLG membrane fared even worse and failed in both scenarios, remaining impermeable to the solutions. This is likely due to its hydrophobic nature resulting from the benzyl pendants in PBLG, resulting in its complete impermeability to aqueous solutions. SEM analysis of the membranes revealed that the PBLG membrane exhibited a very dense appearance, while the UiO-66@PBLG membrane showed a porous architecture where MOF nanoparticles were homogeneously dispersed in the polymer matrix. This porous architecture facilitated solution permeation, allowing for the contact of MOF

particles with small molecules and ions present in the solutions. The UiO-66 + PBLG membrane, despite containing the same amount of MOF within the polymer matrix, was much less permeable to solutions due to the observed phase separation and particle aggregation, significantly reducing the porosity of the membrane matrix. The large areas of PBLG phases hindered solution permeation, and the aggregated UiO-66 particles exhibited reduced surface area, resulting in decreased absorption towards molecules even if the solution passed through. The limited number of channels provided by the aggregated MOF could also get easily blocked in some situations, such as in the case of $K_2Cr_2O_7$ solution filtration. These results demonstrate that MOF-mediated polymerization is a straightforward and useful route toward hybrid materials, providing unique characteristics and explicit superiority compared to hybrid material analogs resulting from simple material mixing in various aspects. Such features can be particularly advantageous for the applications of MOF materials due to their known issues in processability.

## Explorations on the potential applications of MOF@polypeptide MMM materials

As a hybrid material combines attractive features of its constituent components, its usefulness can be diversified easily. Therefore, the above-mentioned MOF@Polypeptide MMM materials are expected to enable potential applications beyond adsorption. While the primary focus of this report is not on this aspect, we believe that a preliminary exploration on the additional features of the MOF@Polypeptide hybrid materials will serve as a source of inspiration and will effectively showcase the advantages of this synthetic approach. Accordingly, we started an initial investigation aiming to evaluate the catalytic capabilities of this MMM material, as catalysis is one of the primary applications for diverse MOF materials. Notably, the UiO-66 nanoparticles employed in NCA polymerization are known to facilitate the hydrolysis of organic phosphates, thus raising the possibility that the catalytic feature can be inherited by the corresponding MMM material. Indeed, by employing a fluorogenic hydrolysis reaction of 4-methylumbelliferone phosphate (4-MUP), we clearly observed the catalytic activity of UiO-66@PBLG for the hydrolysis reaction, as indicated by the appearance of the fluorescent product, 4-methylumbelliferone (4-MU) (Fig. 8a). Clearly, like the molecular absorption capability, the MOF's distinctive functional features can be successfully transferred to the MOF@Polypeptide MMM material, greatly expanding the range of potential applications for the final MMMs.

Furthermore, because of the high tunability of MOF materials used in this strategy, the MOF catalyst for NCA-ROP can be rationally designed or modified, offering the ability to impart specific desirable features to the resulting MMM material without changing the overall polymerization procedure. For example, the introduction of secondary metal catalytic centers into the Zr-MOF framework may facilitate other reactions, while the Zr centers can continue to serve as NCA-ROP catalysts. To demonstrate this possibility, we prepared a new Zr-MOF, $Zr_6O_4(OH)_4(bpydc)_6$ (UiO-67-Py, Supplementary Fig. 28, bpydc = 2,2′-bipyridine-5,5′-dicarboxylate), which is a structural analog of UiO-67 with bipyridines serving as the secondary coordination sites. Copper ions can be incorporated into the framework via the coordination with bipyridine through post-synthetic metalation to generate copper-containing UiO-67 (UiO-67-Cu, Supplementary Fig. 28 and Supplementary Tab. 4)[63], which was further used for the ROP of BLG-NCA. The incorporation of Cu centers endowed the final MMM material (UiO-67-Cu@PBLG) with the capability to mediate copper-catalyzed alkyne-azide cycloaddition (CuAAC) reactions. To confirm this capability, we employed a fluorogenic reaction to monitor the formation of the triazole product, using 3-azidoumbelliferone (3-AU) and 4-ethynylanisole as the substrates. As anticipated, the UiO-67-Cu@PBLG MMM material exhibited catalytic activity for CuAAC, as

evidenced by the distinct cyan fluorescence corresponding to the formation of the click product (Fig. 8b). This compelling demonstration unambiguously highlights the versatility of our synthetic strategy and the modular nature that can be effectively utilized in material design endeavors.

The polypeptide component also plays a crucial role in the functionality of the prepared hybrid materials. In addition to facilitating solvent dispersibility and material processability, the polypeptide component may also contribute to the enhancement of certain features inherited from the MOFs. To demonstrate the possibility, we prepared ZIF-8 nanoparticles (ca. 400 nm in diameter) loaded with 2-methyl-4-chlorophenoxyacetic acid (MCPA), which is a potent and selective phenoxy herbicide widely used worldwide. This created a classic MOF-based sustained release system (Fig. 8c), which could potentially prolong the efficacy of the herbicide in the environment while reducing the overall application dosages required. However, the unmodified ZIF-8 showed a pronounced burst-release profile of MCPA, giving 95% release within 2 days and 99% release within 5 days in PBS (pH = 6). This rapid release behavior is possibly due to the relatively unstable nature of naked ZIF-8 in acidic conditions, and the high water solubility of deprotonated MCPA, which is related to its high mobility in the soil matrix. In contrast, when MCPA-loaded ZIF-8 (MCPA∈ZIF-8) was employed in the polymerization of BLG-NCA to obtain MCPA∈ZIF-8@PBLG, and was subsequently formulated into the corresponding MMM, a significantly improved sustained release profile of MCPA was observed with 72% release over 30 days (Fig. 8c). This substantial enhancement in the sustained release can be possibly attributed to the protection effect conferred by the PBLG layer on the MOFs. Clearly, in this case, the hybridization of MOF and polypeptide not only facilitated the processibility of the MOF materials through membrane preparation, rendering them more amenable to practical applications, but also substantially strengthened the capability of MOF to mediate long-term, controlled release of functional molecules.

## Discussion

In summary, we have demonstrated the validity of a heterogeneous catalyst design for NCA-ROP. The resulting in situ UiO-66 NP-mediated NCA-ROP strategy has shown superior controllability and water-insensitivity. To our knowledge, it is a very rare, if not the only, report of a heterogeneous co-catalytic system used for NCA-ROP. The catalytic system facilitates the polymerization process through a two-step mechanism. The initiation step may involve a coordination-insertion mechanism, wherein the coordinated water molecules on the surface-exposed $Zr_6$ sites of UiO-66 particle facilitate the creation of a COOH-terminated C-terminus of the polypeptide chain, exposing free amine groups for chain propagation. The fast initiation on UiO-66 NPs establishes the well-aligned polypeptide arrays anchored on the MOF surfaces, which results in a proximity-induced acceleration effect for polymerization in the second step. Therefore, the propagation becomes fast and self-sustaining. Importantly, the rational design of separating the initiation and propagation stages has simplified the ways to heterogeneous catalysis, while the benefits of using a heterogeneous system, such as easy product separation and catalyst recyclability, are retained. Moreover, the strategy provides an economical and scalable solution for the preparation of functional polypeptides with good control and a direct route to generating functional polymer-inorganic hybrid materials. The MOF@polypeptide material prepared using our strategy features excellent processability and unique properties as an MMM material when compared to materials derived from pure MOF, pure polypeptide, or a simple mixture of both components. The versatility of this polypeptide-based MMM material preparation method has been demonstrated through several preliminary examples, showcasing a seamless and powerful integration of rationally designed features derived from the constituent components. This innovative approach has enabled diverse applications, including

membrane filtration, catalysis, and sustained release. Overall, our unique catalytic system offers a starting point for bridging the gaps between polypeptide preparation and its practical applications in materials science and engineering.

## Methods

### Synthesis of NCAs

NCAs were prepared using the corresponding α-amino acid with triphosgene, which is a highly toxic reagent. Anyone who works with triphosgene should avoid all personal contact including inhalation, and work only in a well-ventilated fume hood. Always wear protective clothing. The container should be securely sealed after use. All glassware used in the NCA synthesis should be thoroughly dried.

### γ-benzyl-L-glutamate-N-carboxyanhydrides (BLG-NCA) and γ-benzyl-D-glutamate-N-carboxyanhydrides (BDG-NCA)

γ-Benzyl-L-glutamate or γ-Benzyl-D-glutamate (4.74 g, 20.0 mmol) and triphosgene (3.00 g, 10.0 mmol) were added into an oven-dried 100-mL round-bottom flask, into which anhydrous tetrahydrofuran (THF, 40 mL) was then added under nitrogen protection. The mixture was stirred at 50 °C for 3 h under nitrogen. The mixture was cooled down to room temperature, and pure BLG-NCA was obtained by recrystallization in anhydrous THF/hexane three times as a white, needle-like crystalline solid (3.90 g, 74%). The product was stored at −20 °C in the freezer in a jar filled with desiccants. NMR characterization results for D-/L-NCAs were identical in achiral solvents. $^1$H NMR (400 MHz, CDCl$_3$): δ 7.35 (m, 5H), 6.38 (s, 1H), 5.15 (s, 2H), 4.38 (m, 1H), 2.61 (t, J = 6.4 Hz, 2H), 2.31 (m, 1H), 2.14 (m, 1H). $^{13}$C NMR (101 MHz, CDCl$_3$): δ 172.40, 169.06, 152.03, 135.19, 128.73, 128.61, 127.94, 67.12, 56.82, 29.66, 27.04. High-resolution MS (EI, m/z): Calculated for C$_{13}$H$_{14}$NO$_5$ ([M + H]$^+$): 264.0872, found: 264.0874.

### N$^ε$-carboxybenzyl-L-lysine-N-carboxyanhydrides (ZLL-NCA)

N$^ε$-Cbz-L-lysine (2.80 g, 10.0 mmol) and triphosgene (1.50 g, 5.00 mmol) were added into an oven-dried 100-mL round-bottom flask, into which anhydrous THF (30 mL) was added under nitrogen protection. The mixture was stirred at 50 °C for 4 h under nitrogen. The mixture was cooled down to room temperature, and pure ZLL-NCA was obtained by recrystallization in anhydrous THF/hexane four times as a white fluffy solid (1.50 g, 50%). The product was stored at −20 °C in a freezer in a jar filled with desiccants. $^1$H NMR (400 MHz, DMSO-d$_6$): δ 9.08 (s, 1H), 7.31 (m, 5H), 7.24 (m, 1H), 5.01 (s, 2H), 4.42 (m, 1H), 2.97 (m, 2H), 1.68 (m, 2H), 1.25-1.45 (m, 4H). $^{13}$C NMR (101 MHz, CDCl$_3$): δ 170.17, 156.93, 152.56, 136.64, 128.61, 128.25, 128.03, 66.75, 57.48, 40.18, 30.63, 29.14, 21.26. High-resolution MS (EI, m/z): Calculated for C$_{15}$H$_{19}$N$_2$O$_5$ ([M + H]$^+$): 307.1294, found: 307.1299.

### N-carboxyanhydrides-β-benzyl-L-aspartate (BLA-NCA)

L-aspartic acid β-benzyl ester (2.23 g, 10.0 mmol) and triphosgene (1.50 g, 5.00 mmol) were added into an oven-dried 100-mL round-bottom flask, into which anhydrous THF (30 mL) was added under nitrogen protection. The mixture was stirred at 50 °C for 4 h under nitrogen. The mixture was cooled down to room temperature, and pure BLA-NCA was obtained by recrystallization in anhydrous THF/hexane three times as a white needle-like crystalline (1.80 g, 67%). The product was stored at −20 °C in the freezer in a jar filled with desiccants. $^1$H NMR (400 MHz, CDCl$_3$): δ 7.38 (m, 5H), 6.20 (s, 1H), 5.19 (s, 2H), 4.60 (m, 1H), 3.07 (m, 1H), 2.85 (m, 1H). $^{13}$C NMR (101 MHz, CDCl$_3$): δ 169.88, 168.63, 152.02, 135.22, 128.57, 128.50, 128.40, 67.19, 53.60, 35.50. High-resolution MS (EI, m/z): Calculated for C$_{12}$H$_{12}$NO$_5$ ([M + H]$^+$): 250.0710, found: 250.0708.

### Representative protocol of NCA-ROP with UiO-66 NPs

All NCA polymerizations were conducted in a fume hood at room temperature. UiO-66 nanoparticles (3.0 to 30.0 mg, based on the

desired mass ratio of BLG-NCA to UiO-66) were pre-dispersed in DCM (500 μL) by ultrasonic treatment for 40 min, then BLG-NCA (30.0 mg, 0.114 mmol) dissolved in DCM (500 μL) was mixed with the UiO-66 suspension. After vigorously stirring for 20 h (or shorter, based on the desired M/I ratio and conversion), the MOF@Polypeptide was obtained. If free polypeptide was desired, the reaction mixture was centrifuged at 13,523 rcf for 20 min, and the supernatants were added into cold diethyl ether to precipitate the polypeptide product. After centrifugation and drying under vacuum, the obtained PBLG were analyzed by GPC to determine the $M_n$ and dispersity.

Note: Stirring may significantly affect the heterogeneously catalyzed polymerization. For the best result, always place the reaction vial/flask in the center of the stirring plate used.

For BLG-NCA polymerizations with $m_{BLG-NCA}/m_{UiO-66} > 10$, the mass of UiO-66 nanoparticles used was set at 3 mg, whilst more NCA and solvent were used to keep the concentration of BLG-NCA at 0.114 M. Polymerizations of other NCA monomers or in different solvents by different MOF particles were performed similarly.

### Chain-end functionalization
Right after the polymerization, excess 4-nitrophenyl-1-pyrenylmethyl ester (approximately 10 equivalents to PBLG) and 2 μL of triethylamine were added to the reaction, and the mixture was stirred for another 24 h to produce UiO-66@PBLG-Pyrene. The PBLG-Pyrene was isolated and purified by centrifugation and precipitating in MeOH three times.

### UiO-66 NPs recycling
After polymerization, the UiO-66 NPs were collected by centrifugation at 13,523 rcf for 20 min, and then DMF was used as a solvent to wash the UiO-66 NPs twice. The collected UiO-66 NPs were regenerated by placing them in boiling water for 20 min. After cooling down and centrifugation, solvent exchange was done by placing the UiO-66 NPs in acetone for 8 h. The UiO-66 NPs were then washed with acetone three times and dried at 50 °C in a vacuum oven before being used in another polymerization with the same $m_{NCA}/m_{MOF}$ ratio.

## Data availability
The data that support the findings of this study are available within the paper and its Supplementary Information file. Any other data are available from the corresponding authors upon request.

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

## Acknowledgements

Fundings from the National Natural Science Foundation of China (92163127, Y.B.; 21877032, H.X.), Huxiang Youth Talent Support Program of Hunan Province (2022RC1107, Y.B.; 2022RC3046, H.X.), and Hunan Provincial Science & Technology Department (2022SK2003, 2022JJ10007, H.X.) are sincerely acknowledged. The authors thank Prof. Jianjun Cheng of Westlake University for his suggestions on this project.

## Author contributions

Y.L., Z.R., H.X., and Y.B. conceived and designed the experiments. Y.L., Z.R., N.Z., X.Y., and Z.C. performed the experiments. Y.L., Z.R., Q.W., H.X., and Y.B. analyzed the data. Y.L., Z.R., H.X., and Y.B. prepared the manuscript with contributions from all authors. Y.L. and Z.R. contributed equally to this work.

## Competing interests

The authors declare no competing interests.
