## [Peer Review File · Nature Communications]

A Nanoscale MOF-Based Heterogeneous Catalytic System for the Polymerization of N-Carboxyanhydrides Enables Direct Routes toward Both Polypeptides and Related Hybrid MaterialsReviewers' Comments:

Reviewer #1:

Remarks to the Author:

This communication reports a NCA polymerization initiated from MOF particle surface. The key novelty is it cleverly utilised water or alcohol complexed at MOF surface as initiation sites for NCA-ROP and the formed surface polypeptides can be easily separated via simple centrifugation. The recovered MOF can be recycled. The observed accelerated chain propagation is known in the literature due to the close arranged brush structure in alpha helix form. The formed polypeptides are in good low polydispersity without any other mediating group used and the molecular weight can be achieved to as high as 140K. I believe this work teach new knowledge in NCA polymerization and provided new a tool for surface modification in biological applications. Therefore, I support this paper to be published in Nature Communication.

Minor points:

The paper says this is a two-stage polymerization and I am not sure if such term is necessary. To me this is one process which may involve different propagation rate stage due to the formed alpha helix structure. Also, Fig 3b is not clear as chain propagation with auto catalysis should resulted in much longer brush.

For Fig 3c, should the accelerated NCA-ROP have resulted in polymers with only 5-10 DP?

Greg Qiao

Reviewer #2:

Remarks to the Author:

This manuscript reports a MOF-supported surface initiation system (based on UiO-66 NP) for ring-opening polymerization of amino-acid N-carboxyanhydrides. This heterogeneous catalysis system offers, in principle, good control and insensitivity to water. It enhances the initiation step via surface-anchored metal species while the propagation is enhanced by already described alpha helix cooperativity (Nat. Chem., 2017, 9, 614). Once synthesized, the polymers can be cleaved from the surface for recovery and the heterogeneous catalyst reused (even though these last data are not fully convincing). Despite some interesting observations, in my opinion, this article is not suitable for publication since it presents 3 major shortcomings: i) the first issue is that many discussions and assertions are not based on the experimental facts presented (they are at best hypotheses); ii) the second issue is related to the bibliographic context which completely forgets that the use of a heterogeneous process including surface-initiated ROP has already been widely described (Chem. Commun., 2014, 50, 4971 for a review and Angew. Chem., Int. Ed., 2003, 42, 1839 for a representative article); iii) the last issue is that the article is hyped (misrepresented) while it does not develop a very original approach with respect to the state of the art (among others: Angew. Chem. Int. Ed., 2021, 60(7): 3449, ref 43 of the manuscript). At best, the manuscript publication is premature (see the nanocomposite part that is too scarce, as compared to Nat. Commun. 2020, 11, 2051), at worst, this work should be published in a more specialized journal (the other part of the manuscript). The details of the comments and observations on the article are described below:

Introduction: The introduction does not present the context of surface-initiated ROP of NCA in any way (Chem. Commun., 2014, 50, 4971), which is an already abundant literature including from nanomaterial surfaces. In this context, figure 1 is particularly unrepresentative: it is placed in a

context of work developed by a single group (J.J. Cheng, see the corresponding refs 11, 13, 14 and 15 of the manuscript) and forgets many old or recent elements showing the advantage of ROP of NCA in a multiphase context (for recent examples: ROP in Emulsion, polyHIPE, ROPISA etc.). This includes the cooperative effect discussed in the manuscript. The introduction and part 1 should be fully revised to take into account all these comments.

Main result (figure 2): The breakthrough with the state of the art is not obvious (see ref 43 of the manuscript) and several questions arise with respect to the result presented in figure 2. Are MOF pure in solution? Are they not mixed with small clusters or metal salts in solution (leaching)? Controls with these clusters and with these species ($ZrOCl_2$) in solution are necessary to convince of the result presented by the authors. Also, this question must be seriously considered in view of the procedures implemented (use of ultrasonication that can degrade the metallic structures). On another hand, the results on Cr^{3+} and on Cu^{2+} should be added to the esi part. Also, the NCA concentrations used in DCM are high and in some cases an increase in viscosity is expected (Colloid. Polym. Sci. 2013, 291:1353 and H. Schlaad related works). This may explain the turbidity discussed fig 2a. Finally, why using HF? Are there simpler methods to detach the polypeptide? This treatment is not attractive in a context where its use has almost completely disappeared from small peptide synthesis. Is this treatment (and others presented in the article) consistent with the ester moiety present on the amino acid side chain?

Figure 3. There is limited mechanistic data and the MALDI analyses do not support the proposed depicted mechanism. This mechanism has no value (and is certainly obvious), it is totally hypothetical. Free Zr species could also catalyze the process and the polypeptide remained attached via non-covalent linkages (or organometallic bonding). The proposed Zr-O bonding could be displaced by simply adding more Zr species (instead of HF). Concerning MALDI analyses, why give only small masses? These data are not very convincing. The whole spectrum should be non-zoomed to masses in agreement with those obtained by NMR or SEC. Figure 3a, unassigned peaks correspond to a chain end cyclization due to temperature (cf Polym. Chem., 2010, 1, 514). In figure S5: what was the objective? Why using MALDI (at such masses ESI is generally used instead)? Below 1000, MALDI is not precise enough ?

Other comments regarding discussion:

- An interesting scientific question coming from the proof of concept developed in this work is the importance of the grafting density (water content) as opposed to the strength of secondary structuring. This is not currently well addressed. For instance, fig. S6, why does the D,L copolymerization also have 2 slopes? In this same figure, the first slope is difficult to distinguish (fig 6a).
- An important hypothesis could be that MOF only serves to make water soluble in the DCM organic phase. Water and organic solvent are not miscible, so there is not much interest to add water? The word "hydrous DCM" is rather incorrect. Controlling the amount of water probably controls the surface initiation density.
- Are MOFs soluble in an aqueous phase, does this influence the possible release of metallic species?
- Figure S7: water content could also explain the differences observed (For THF and DMF, water is miscible...).
- Figure 5: Data on the reuse of MOFs is somewhat unconvincing. How can the differences be explained? Could the amount of water on the surface of the MOFs be the cause of the observed differences?
- The part describing the hybrid materials is very interesting but does not correspond to the same narrative as the one presented before in the manuscript. This part should be removed from the manuscript.

Experimental data: At the end of the centrifugation, the authors should discuss the yield of MOFs (and associated polypeptides) that have been recovered. Table 1: two significant digits after the decimal point are sufficient for the dispersity.

Manuscript format: The article is not necessarily well written and it contains several typos. Ex: Figure 4d should be 4e, « solvent exchange was by palce », etc...

Reviewer #3:

Remarks to the Author:

In this manuscript, Liu et al. reported a water-initiated NCA polymerization catalyzed by MOF nanoparticles, in which MOF acts as a heterogeneous catalyst and controls molecular weight in the chain initiation stage, generating helical peptide arrays with a self-accelerating chain propagation process. The MOF nanoparticles can be recycled by separation and can also be used directly to prepare hybrid materials. However, the self-accelerating effect of helical polypeptides in chain propagation stage has been well recognized, and the ability of Lewis acidic metal sites to activate carbonyl groups is also well documented in literature. Therefore, there is not much surprise as a polymerization method in this study, although a detailed study was carried out. In addition, my interest on this study is diminished by some major defects about this polymerization strategy that is unlikely to be solved with further optimization. The finding in this study doesn't meet the high standard for publication in Nature Communications.

Some detailed concerns and questions.

1. In this paper, MOF catalyzes water or ethanol to initiate NCA interfacial polymerization and obtain the self-accelerating effect during the chain propagation stage. What are the advantages and obvious breakthrough of this paper compared with existing fast polymerization systems?
2. It is not surprising the NCA monomers will decompose when polymerization is carried out over 8h in an aqueous condition in this study. The decomposition products can also initiate NCA polymerization and result in a complex and uncontrollable result. Some detailed studies are expected on these issues.
3. As reported in this study the racemic NCA monomer (D,L-configuration) undergoes polymerization without the self-accelerating cooperative effect, and the monomer conversion is only 68%. For such an incomplete polymerization, how to control the ratio of different subunits in the obtained polymer?
4. In this study, only N ϵ -carboxybenzyl-L-lysine (ZLL) NCA and γ -benzyl-glutamate NCA in L and/or D configurations are used as the monomer for polymerization. It is important to demonstrate that this polymerization method is suitable to other common NCA monomers by incorporating other types of monomers for polymerization.
5. To demonstrate the usefulness, it is necessary to study the copolymerization and block polymerization of this polymerization method.
6. The authors propose that MOF-catalyzed NCA polymerization provides a direct route for the preparation of MOF-polypeptide hybrid materials. However, the advantages of this modified MOF and its potential applications in materials science are not explored or conceptually demonstrated in this study.
7. Regarding the catalyst stability, whether this MOF surface modification exists stably?
8. According to the description in this study, for this polymerization method MOF acts not only as a catalyst but also as a co-initiator to control the molecular weight of the polymerization product. More demonstration on the control of variable chain length (both shorter and longer chain length) should be studied in Table 1.
9. The authors mention other types of autocatalytic macromolecular initiators, so the cost of preparing MOF in this paper should be compared with other initiators cited by in this manuscript.
10. The authors propose that the MOF system can be cycled, providing polymerization results after

three cycles. However, the molecular weight of the obtained polymers after cycling increase significantly each time, indicating the loss of activation sites. Whether the MOF catalyst fails largely after more cycles? How will the recycled MOF catalyst control the molecular weight of the polymers?

11. Because the MOF catalyst and the polypeptide can be separated by centrifugation and acidification, it is a concern that during polymerization the polypeptides be partially detached from the MOF catalyst and result in both defect in the catalyst surface and soluble polypeptide in the solution. It is likely the defect on the catalyst surface can diminish the acceleration effect in polymerization. The dissolved polypeptide in solution will continue the polymerization process but with different speed compared to that on the MOF surface. How will these affect the controllability of this polymerization method?

12. As a related question, what is the percentage of polypeptide detachment from the surface of catalyst during polymerization? The polymer fraction in Fig 2a after centrifugation shows some turbidity, which indicates possible residual catalyst component in the polymer product.

13. Other minor issues that need to be explained

a) The authors should explain the reason for the shoulder peaks in GPC in Fig 2c.

b) All MALDI peaks in Fig 3a and 3c should be assigned.

c) In addition to MALDI, the polymerization kinetics of ethanol-initiated and water-initiated should be compared.

Response to Reviewers

Reviewer 1:

Minor points:

This communication reports a NCA polymerization initiated from MOF particle surface. The key novelty is it cleverly utilized water or alcohol complexed at MOF surface as initiation sites for NCA-ROP and the formed surface polypeptides can be easily separated via simple centrifugation. The recovered MOF can be recycled. The observed accelerated chain propagation is known in the literature due to the close arranged brush structure in alpha helix form. The formed polypeptides are in good low polydispersity without any other mediating group used and the molecular weight can be achieved to as high as 140K. I believe this work teach new knowledge in NCA polymerization and provided new a tool for surface modification in biological applications. Therefore, I support this paper to be published in Nature Communication.

We thank the reviewer for the encouraging comments.

The paper says this is a two-stage polymerization and I am not sure if such term is necessary. To me this is one process which may involve different propagation rate stage due to the formed alpha helix structure. Also, Fig 3b is not clear as chain propagation with auto catalysis should resulted in much longer brush.

We agree with the reviewer that the whole polymerization could be considered as one process as the chain propagation mechanism remains unchanged. In the kinetic analysis, we described it as "two-stage" simply based on the different polymerization rates (as seen in Figure 4a and 4b), and it is a term we borrowed from Prof. Cheng as he and his colleagues used this term to describe the rate change in the self-accelerated NCA-ROPs.

However, we realize that our description may be confusing as the MOF is working on the catalyzed initiation, and this is actually a different "step" in the whole polymerization because it is involving OH attacking the NCA carbonyl, but in the chain propagation (not directly catalyzed by MOF) it is the amine that attacks NCA carbonyl. We have also used "two-stage" in this part of description and it can be quite misleading. Following the reviewer's comments, in this revised manuscript we have limited the use of the term "two-stage" to the kinetic analysis part, keeping the term consistent with previous works. In the descriptions involving initiation and polymerization mechanisms, we simply use the words "initiation" and "propagation" to differentiate the two "steps". We think that this will better describe the mechanistic and kinetic features of this self-acceleration without leading to confusion or misunderstanding.

In addition, Figure 3b is now revised according to the reviewer's comment. The brushes in the cartoon picture have their length increased to make the propagation in the accelerated stage more distinguishable.

For Fig 3c, should the accelerated NCA-ROP have resulted in polymers with only 5-10 DP?

We apologize for causing misunderstanding in the way we present the data in Figure 3c (now Figure 4b). No, normally the accelerated NCA-ROP will result in polymers with high DPs (can be ≥ 1000). However, for the experiment here, the main purpose was to reveal the initiation mechanism by doing end-group analysis with MALDI-TOF. Since it is inherently quite difficult for those long polypeptides to give well-defined sharp peaks on MALDI-TOF MS, in this experiment we were analyzing the “infant” chains, i.e., just-initiated chains that were short and were yet to enter the self-accelerated ROP stage. Because the chain-end’s structure is not affected by variations in chain length, by “seeing” the very short chains we were able to get precise m/z ratios of the chains and to decipher the chain-end’s structure. As one can see in the figure, we could confirm the active initiating groups in this system were water or ethanol coordinated on the NPs’ surfaces as displayed in Figure 4a and 4b, respectively. These results again supported the mechanism we proposed.

In this revised manuscript, we have made changes to Figure 4a, 4b, their captions, and some related sentences in the article, indicating that these analytes were obtained by quenching the polymerizations shortly after initiation. We hope that these changes will better explain the experiments’ design and aim.

Reviewer 2:

This manuscript reports a MOF-supported surface initiation system (based on UiO-66 NP) for ring-opening polymerization of amino-acid N-carboxyanhydrides. This heterogeneous catalysis system offers, in principle, good control and insensitivity to water. It enhances the initiation step via surface-anchored metal species while the propagation is enhanced by already described alpha helix cooperativity (Nat. Chem., 2017, 9, 614). Once synthesized, the polymers can be cleaved from the surface for recovery and the heterogeneous catalyst reused (even though these last data are not fully convincing). Despite some interesting observations, in my opinion, this article is not suitable for publication since it presents 3 major shortcomings: i) the first issue is that many discussions and assertions are not based on the experimental facts presented (they are at best hypotheses); ii) the second issue is related to the bibliographic context which completely forgets that the use of a heterogeneous process including surface-initiated ROP has already been widely described (Chem. Commun., 2014, 50, 4971 for a review and Angew. Chem., Int. Ed., 2003, 42, 1839 for a representative article); iii) the last issue is that the article is hyped (misrepresented) while it does not develop a very original approach with respect to the state of the art (among others: Angew. Chem. Int. Ed., 2021, 60(7): 3449, ref 43 of the manuscript). At best, the manuscript publication is premature (see the nanocomposite part that is too scarce, as compared to Nat. Commun. 2020, 11, 2051), at worst, this work should be published in a more specialized journal (the other part of the manuscript).

We thank the reviewer for these constructive comments. For the 3 major shortcomings mentioned by the reviewer, our responses are the following:

i) the first issue is that many discussions and assertions are not based on the experimental facts presented (they are at best hypotheses)

We assume that the reviewer here means the issues he/she had stated in the later parts of the review. These issues were addressed in this revised manuscript. Please see our point-to-point response below.

ii) the second issue is related to the bibliographic context which completely forgets that the use of a heterogeneous process including surface-initiated ROP has already been widely described (Chem. Commun., 2014, 50, 4971 for a review and Angew. Chem., Int. Ed., 2003, 42, 1839 for a representative article)

We have to admit that we did not cite literature reports on this topic, but we did not forget them. We wrote the manuscript in this way because we think that the main idea of our work is not a revisit of surface-initiated ROP. We are aware of the existing reports on heterogeneous, surface-initiated ROP (grafting-from), but despite that our ROP is also initiated on MOF surface, it was initiated by the surface absorbed water instead of functional groups covalently tethered on the surface. Thus, the whole preparation process is more a catalytic one in nature, as the polypeptides formed are not covalently attached to the surface and can be easily detached from the MOF particles. The surface is playing a catalyst-typed role: the initiators are absorbed on the surface and reacting with the monomers, and the surface does not form irreversible covalent connections to the resulting polymer chain. For all the surface-initiated ROP reported before, the mechanism is fundamentally different: the surfaces were initiators, and polymers were rooting on the surface

and growing from there. Surface-initiated ROPs cannot be used for the preparation of free polypeptides, at least not easily.

From the rest of the reviewer's comments, we think that there is a huge misunderstanding – the reviewer may not have noticed that the polypeptide chains were very loosely connected to the MOF particles, as he/she asked “why using HF? Are there simpler methods to detach the polypeptide?” As this is one of the most important novelties we wanted to present with the article, we hope the reviewer can re-evaluate our manuscript with the consideration of such features of the ROP we present. Also, we realize that we may not have well described our experimental results and related mechanistic analysis, which had led to the misunderstanding. We apologize for that, and hopefully with the changes we made in this revision, the idea of the manuscript can be more reader-friendly.

Finally, based on the reviewer's comment, now we do feel that some mentioning on surface-initiated NCA-ROP will be helpful. In this revised manuscript, we have re-written a large portion of the introduction part, and have now included citations on some notable works involving surface-initiated NCA-ROP, and the “grafting-on” of polypeptide onto materials surfaces. By having these citations mentioned, readers can make comparisons between ours and others' works so that they may see the difference in the fundamental design philosophy.

iii) the last issue is that the article is hyped (misrepresented) while it does not develop a very original approach with respect to the state of the art (among others: Angew. Chem. Int. Ed., 2021, 60(7): 3449, ref 43 of the manuscript).

As we have stated in our response to ii), the “hyped” feeling was likely from the misunderstanding caused by our writing. Again, we are not presenting another surface-initiated NCA-ROP, and we hope the reviewer can re-evaluate the originality of the article as we are making ourselves clearer with all the changes in this round. Also, in this revision we have greatly enriched the nanocomposite part. We hope the reviewer can also find these new data interesting.

The details of the comments and observations on the article are described below:

Introduction: The introduction does not present the context of surface-initiated ROP of NCA in any way (Chem. Commun., 2014, 50, 4971), which is an already abundant literature including from nanomaterial surfaces. In this context, figure 1 is particularly unrepresentative: it is placed in a context of work developed by a single group (J.J. Cheng, see the corresponding refs 11, 13, 14 and 15 of the manuscript) and forgets many old or recent elements showing the advantage of ROP of NCA in a multiphase context (for recent examples: ROP in Emulsion, polyHIPE, ROPISA etc.). This includes the cooperative effect discussed in the manuscript. The introduction and part 1 should be fully revised to take into account all these comments.

We thank the reviewer for this comment. We think the issue here is miscommunication rather than not being representative. The central idea of this article is “catalysis”, therefore we did not include too much discussions on surface-initiated NCA-ROP. A heterogeneous (surface-initiated) NCA-ROP does not indicate that the system is catalyzed with a heterogeneous catalyst, and these are fundamentally different concepts. For example, emulsion polymerization technique is a heterogeneous polymerization commonly used for free radical polymerizations and it will yield

latex. It can be difficult to completely remove the surfactants from a latex to give pure polymers. In contrast, in a polymerization mediated by a heterogeneous catalyst, like Ziegler-Natta catalyst-mediated polyolefin synthesis, the catalyst will be in a different phase and can be very easily separated from the polymer product. Please note that we are reporting a new heterogeneous catalyst, not a new heterogeneous polymerization or a surface-initiated polymerization.

However, we do want to take the criticism from the reviewer. If we failed to highlight the advantage and unique features of our system with all the text and figures, that was our problem. We have therefore revised the introduction part and the first figure, so that they can more explicitly indicate the nature of this system (and our aim): a heterogeneous catalytic system that can be used for the synthesis of both free polypeptides and hybrid materials. We have also added several citations related to polypeptide-based hybrid materials, prepared using grafting-from (surface initiation), grafting-to or self-assembly methods. We hope that by citing these literatures, the readers may better see the difference between previous works and our system.

Again, with this revised manuscript, we hope that the reviewer will find the catalytic polymerization process interesting, and re-evaluate the novelty of our work. We admit that there have been tremendous amount of literatures talking about surface polymerizations or heterogeneous polymerizations, but they are not something we wanted to discuss too much in this manuscript, as they are actually not directly related to catalysis. However, as we have stated above, representative works in these categories are cited to allow comparisons.

Main result

Figure 2: The breakthrough with the state of the art is not obvious (see ref 43 of the manuscript) and several questions arise with respect to the result presented in figure 2. Are MOF pure in solution? Are they not mixed with small clusters or metal salts in solution (leaching)? Controls with these clusters and with these species ($ZrOCl_2$) in solution are necessary to convince of the result presented by the authors. Also, this question must be seriously considered in view of the procedures implemented (use of ultrasonication that can degrade the metallic structures).

We thank the reviewer for those comments.

For the breakthrough part, please refer to our responses above. As our system is a catalytic rather than a surface-initiating system, they are not quite comparable. There has never been any reported heterogeneous catalyst for NCA-ROP, and ours is the first.

Also, we appreciate these concerns. However, UiO-66 is a very classic type of MOF and its preparation protocol is highly reliable and reproducible. In our work, we were following existing standard procedures for the MOF part, and their purity should be assured. The preparation of a MOF is, in nature, a recrystallization process, and the material (basically inorganic salt crystals) yielded is always thoroughly washed. It is not likely that the MOF used in the experiment is contaminated with other metal species.

For the leaching part, we are sure that there are no small clusters or salts that can affect the NCA-ROP. The total insolubility of UiO-66 in organic solvents like chloroform is well known, and ultrasonication is a standard protocol for nano-MOF preparation. However, to give direct and most assuring evidence, we have performed the experiments suggested by the reviewer. We

tested the leaching by putting 30 mg of UiO-66 nanoparticles in 1 mL of anhydrous DCM, or 1 mL of 10% H₂O/DCM mixture, and the mixtures were ultrasonicated for 1 h followed by vigorous stirring for another 24 h. After centrifugating at 12000 rpm for 30 min, the collected supernatants were evaporated and prepared for ICP-MS analysis to detect the amount of leaked Zr species. The ICP-MS result showed that there was almost no Zr species in the supernatant in either case (0.19 ppb of Zr in anhydrous DCM, 0.25 ppb of Zr in 10%H₂O/DCM mixture), which indicated that the UiO-66 nanoparticles were very stable during polymerization. The protocol and the results are now provided in the Supporting Information (protocol section and Table S1). In addition, we tried polymerization with ZrOCl₂, and it revealed that the NCA-ROP could not be initiated by ZrOCl₂, as shown below. No polypeptide product was detected by FT-IR spectroscopy even after 24 h. This result was added into the revised Supporting Information (Figure S6).

On another hand, the results on Cr³⁺ and on Cu²⁺ should be added to the esi part.

We thank the reviewer for the suggestion, and the polymerization results of MIL-101 (Cr³⁺) and HKUST-1 (Cu²⁺) nanoparticles were displayed below. These results have been added into the Supporting Information as Figure S2.

Figure S2. The results of BLG-NCA polymerization initiated by various MOF nanoparticles at $m_{\text{BLG-NCA}}/m_{\text{MOF}} = 5$, $[\text{BLG-NCA}]_0 = 0.114 \text{ M}$. (a) FT-IR spectra of BLG-NCA monomer and the reaction mixtures after 12 h of NCA-ROP initiation attempt by various MOF nanoparticles. (b) GPC curve overlay of the produced PBLG by UiO-67, ZIF-8 and MIL-101 nanoparticles, respectively.

Also, the NCA concentrations used in DCM are high and in some cases an increase in viscosity is expected (Colloid. Polym. Sci. 2013, 291:1353 and H. Schlaad related works). This may explain the turbidity discussed in fig 2a.

The NCA concentration used in DCM is 0.114 M, and at this concentration, all the NCA monomers were dissolved quite well with clear and transparent appearances. Even at a very high concentration of 0.4 M there is no turbidity for BLG-NCA solutions. The turbidity in Figure 2a was caused by the colloid of PBLG-coated MOF nanoparticles in organic solvent. We have updated the figure (now Figure 3a) with new photos to better show the appearance changes of mixtures before and after polymerization. Especially, picture of BLG-NCA solution is provided to show that the turbidity was not caused by the monomer.

Finally, why using HF? Are there simpler methods to detach the polypeptide? This treatment is not attractive in a context where its use has almost completely disappeared from small peptide

synthesis. Is this treatment (and others presented in the article) consistent with the ester moiety present on the amino acid side chain?

We apologize for such misunderstandings. We are reporting a catalytic process, therefore the polypeptide product must not be tightly attached to the MOF (the catalyst). Actually, as we had stated in the manuscript, centrifugation was simply employed to remove UiO-66 from the polypeptide product. The use of HF for MOF digestion was not the typical treatment in the manuscript to collect polypeptides from the MOF nanoparticles – it was used as a control to show that the polymer product isolated by centrifugation was identical to that from digestion (complete dissolution of MOF).

We have made changes to the related figures (especially Figure 2b) and texts to better indicate that centrifugation was the primary way of collecting the polymer products.

Figure 3: There is limited mechanistic data and the MALDI analyses do not support the proposed depicted mechanism. This mechanism has no value (and is certainly obvious), it is totally hypothetical. Free Zr species could also catalyze the process and the polypeptide remained attached via non-covalent linkages (or organometallic bonding).

As we have stated in the response above, there was no free Zr species in the solution as revealed by ICP-MS, and $ZrOCl_2$ could not initiate NCA polymerization.

We do not understand the other parts of the comment here. Why is the MALDI analysis not supporting the mechanism? It clearly showed the end-group is determined by the external initiator (H_2O or ethanol), therefore it is a catalytic process rather than a traditional surface-initiated polymerization. The surface does not initiate the polymerization directly, but provides anchors for the monomer and the initiator and catalyzes the initiation process.

The proposed Zr-O bonding could be displaced by simply adding more Zr species (instead of HF).

We thank the reviewer for this suggestion, but we did not use HF for polymer detachment. This was clearly stated in the manuscript and is more convenient than adding more Zr salt.

Concerning MALDI analyses, why give only small masses? These data are not very convincing. The whole spectrum should be non-zoomed to masses in agreement with those obtained by NMR or SEC.

We thank the reviewer for this comment. However, this was not a zoomed spectrum. Since we were exploring the initiation mechanism, we force-stopped the polymerization shortly after the initiation to avoid the formation of long polypeptides. This was done because for the measurement of MALDI-TOF with polymer samples, shorter chains are much more easily detected and the m/z values could be more accurate. In fact, it is very difficult to identify long polypeptides with MALDI-TOF, and the mass change caused by end-group difference is too small when the chain is too long. Therefore, it was not necessary to bother with long polypeptides just for end-group analysis (end groups never change during chain propagation). These MALDI MS spectra were not, and should not be in agreement with those obtained by NMR or SEC, because the samples were prepared in different ways.

Figure 3a, unassigned peaks correspond to a chain end cyclization due to temperature (cf Polym. Chem., 2010, 1, 514).

These unassigned peaks are apparently minor peaks. Despite being a “soft” ionization technique, MALDI-TOF analysis still leads to some fragmentations and other reactions, including cyclization. The analysis should be focused on the major peak series.

The updated Figure S7 and S8 now provide more detailed peak assignment information.

In figure S5: what was the objective? Why using MALDI (at such masses ESI is generally used instead)? Below 1000, MALDI is not precise enough?

This figure is the zoomed part of the MALDI-TOF spectra that used for end group analysis, which showed the existence of hydrolysis intermediate during the polymerization of low molecular weight of PBLG.

This figure and related descriptions are removed in this round of revision, because we think that the characterization cannot rule out the possibility of NCA hydrolysis during sample preparation and testing (which also generates benzyl glutamate).

Other comments regarding discussion:

- An interesting scientific question coming from the proof of concept developed in this work is the importance of the grafting density (water content) as opposed to the strength of secondary structuring. This is not currently well addressed. For instance, fig. S6, why does the D,L copolymerization also have 2 slopes? In this same figure, the first slope is difficult to distinguish (fig 6a).

We thank the reviewer for this comment. The reviewer is correct on the effect of the grafting density. In fact, the grafting density's effect on the extent of NCA-ROP self-acceleration is well documented (*Nat. Chem.* 2017, 9, 614-622). As we have shown in Figure S11, S12, the property (surface water content) of the MOF nanoparticles was stable enough to allow reproducible self-acceleration effect and molecular weight control. And in Figure S14, we changed the water content to check the effect on polymerization rate and molecular weight. Therefore, we consider it an already-addressed issue.

As for the copolymerization of D,L-monomer, the two slopes were caused by the change in system homogeneity. In the beginning of the polymerization, the MOF nanoparticles are naked and do not well disperse in the organic solvent, and at this stage the polymerization is slower. As chain propagation continues, the MOF particles become gradually more well-coated and form more stable colloid in the solvent, thus it will be easier for the monomer (dissolved in the organic solvent) to react with the chain-end on the MOF particles, increasing reaction rate.

We are not sure if the reviewer is question us on the reproducibility by comparing Figure S6a and S6b (now Figure S9a, S9b). If he/she is, we want to remind that Figure S9a is drawn in linear scale while Figure S9b is drawn in log scale. Therefore, the slopes in these figures have different physical meanings and should not be compared. In fact, Figure S9b is a re-plotted figure of Figure S9a using the same data sets. They do not represent any inconsistency in experiment results.

- An important hypothesis could be that MOF only serves to make water soluble in the DCM organic phase. Water and organic solvent are not miscible, so there is not much interest to add water? The word “hydrous DCM” is rather incorrect. Controlling the amount of water probably controls the surface initiation density.

With respect, we disagree with the reviewer on this point. Water and dichloromethane are indeed not miscible, but their solubility in each other is not negligible. At room temperature, DCM’s solubility in water is approximately 2 g per 100 g, and water’s solubility in DCM is approximately 0.25 g per 100 g (0.03 M). Clearly, such solubility may lead to significant effect in NCA-ROP and other organic reactions, and this is why drying dichloromethane is a common organic chemistry practice. And, since bench-top (not dried) dichloromethane is not saturated with water, adding extra water will certainly increase the water content to the maximum in the organic phase, making changes to the molecular weight of the final polymer. This is related to the previous question from the reviewer: the impact of the water content. Controlling the amount of water indeed controls the surface initiation density, to a maximum as limited by water’s solubility in DCM and the MOF nanoparticle’s total surface area. All these data are actually in the manuscript (Figure S11, S14) – the reviewer had probably missed them.

To avoid confusion, we have adopted the reviewer’s comment and changed the term “hydrous DCM” to “bench-top DCM (not dried)” in the revised manuscript.

- Are MOFs soluble in an aqueous phase, does this influence the possible release of metallic species?

No, UiO-66 nanoparticles are neither soluble in water nor in organic solvents. As we have stated in our response above, we have demonstrated with ICP-MS that no metallic species were released in the polymerization condition.

- Figure S7: water content could also explain the differences observed (For THF and DMF, water is miscible...).

As we have stated in the figure caption, all these solvents were thoroughly dried. Therefore, the water miscibility of these solvents is an irrelevant factor here.

The better self-acceleration effect of NCA-ROP in chlorinated solvent is also documented, and our result is consistent with these reports (*Nat. Chem.* 2017, **9**, 614-622 and *J. Am. Chem. Soc.* 2019, **141**, 8680–8683).

- Figure 5: Data on the reuse of MOFs is somewhat unconvincing. How can the differences be explained? Could the amount of water on the surface of the MOFs be the cause of the observed differences?

Yes, as we have already explained in the manuscript, the increasing molecular weight as the MOF nanoparticles were recycled for more times was caused by the loss of water coordinating sites on the nanoparticles’ surfaces (they may be taken by short oligopeptide chains sticking on the MOF surface). Despite the reduced binding sites, the recycled catalyst still offers good control over molecular weight and dispersity with different $m_{\text{NCA}}/m_{\text{MOF}}$ ratios (Figure S18).

- The part describing the hybrid materials is very interesting but does not correspond to the same narrative as the one presented before in the manuscript. This part should be removed from the manuscript.

We thank the reviewer for this suggestion. We think that the very reason that makes the reviewer feeling that way is miscommunication, as he/she did not notice that the polymerization is actually versatile (he/she apparently thought that we were reporting a surface-initiated polymerization and the polymers must be cut off by HF to complete the synthetic story). In fact, our MOF-based strategy offers straightforward ways in either getting pure polypeptide or preparing hybrid materials, and the two together combine into a complete synthetic story.

Considering that this reviewer is finding the hybrid material interesting, and reviewer #3 is questioning us on the lack of application demonstration, we have added a proof-of-concept study using mixed-matrix membranes prepared from our MOF-polypeptide hybrid material. We have also changed the writing of the manuscript to make the two parts of the article more seamlessly integrated, so that readers will not find the later part in a different narrative as the first half is.

Experimental data: At the end of the centrifugation, the authors should discuss the yield of MOFs (and associated polypeptides) that have been recovered. Table 1: two significant digits after the decimal point are sufficient for the dispersity.

Thanks for these suggestions. The recovery yield of MOF nanoparticles was about 87% after centrifugation, and the isolated yields of polypeptides after centrifugation and precipitated in ether was ranging from 54% to 93% (like most polymerization techniques, low molecular weight polymers give lower yields). We have added discussions on the yields of MOF nanoparticles and polypeptides into the manuscript. Also, the significant digits of the dispersity values have been changed as suggested in Table 1.

Manuscript format: The article is not necessarily well written and it contains several typos. Ex: Figure 4d should be 4e, « solvent exchange was by palce », etc...

We thank the reviewer for this comment. In addition to the typos, we realized that many of the writings were not easily understandable or even misleading. To address these issues, we have revised the whole manuscript. We hope that the article is now in a better state for publication.

Reviewer #3:

In this manuscript, Liu et al. reported a water-initiated NCA polymerization catalyzed by MOF nanoparticles, in which MOF acts as a heterogeneous catalyst and controls molecular weight in the chain initiation stage, generating helical peptide arrays with a self-accelerating chain propagation process. The MOF nanoparticles can be recycled by separation and can also be used directly to prepare hybrid materials. However, the self-accelerating effect of helical polypeptides in chain propagation stage has been well recognized, and the ability of Lewis acidic metal sites to activate carbonyl groups is also well documented in literature. Therefore, there is not much surprise as a polymerization method in this study, although a detailed study was carried out.

We thank the reviewer for this comment. We agree with the reviewer that the self-accelerating effect of helical polypeptides has been well recognized, and the ability of Lewis acidic metal sites to activate carbonyl groups is also well documented in literature. However, the combination of the two as a low-risk strategy to prepare heterogeneous catalyst is unprecedented. As long as a strategy is solving real important issues in the field, we consider a “not surprising design” an elegant one, and such simple systems can offer advantages in cost and reproducibility. Also, as we have stated in the manuscript (and also revealed in some newly added controls), it is not that simple to get a working metal catalyst. Not all the MOFs and metal salts are working well for NCA-ROP catalytic purposes. Importantly, as our revised title is suggesting, the polymerization method offers a direct route toward both polypeptide and related hybrid materials. Such versatility brings the strategy great value in addition to its advantages in cost-effectiveness, efficiency and low environment susceptibility.

In addition, my interest on this study is diminished by some major defects about this polymerization strategy that is unlikely to be solved with further optimization. The finding in this study doesn't meet the high standard for publication in Nature Communications.

We assume that those “major defects” were listed in the rest part of this reviewer's comments. These concerns and questions have been carefully addressed. Please see our point-to-point response below.

Some detailed concerns and questions.

1. In this paper, MOF catalyzes water or ethanol to initiate NCA interfacial polymerization and obtain the self-accelerating effect during the chain propagation stage. What are the advantages and obvious breakthrough of this paper compared with existing fast polymerization systems?

We thank the viewer for this comment. Indeed, in the previous submission, we have mostly failed to highlight the advantage of our strategy. The manuscript is now fully revised for a better presentation.

Existing fast polymerization systems mainly use special macromolecular scaffolds and small molecular initiators. The cost of these systems can be an issue, especially for those macromolecular scaffolds, but more importantly, they can only be used to prepare free polypeptides (and for some of them mandatory tags like oligoPEG are installed upon polymerization). These systems actually offer very limited aid in the preparation of polypeptide-

based hybrid materials: polypeptide in such hybrid materials still need to be introduced through difficult material-polypeptide conjugation process (grafting-to), or traditional surface-amine-initiated NCA-ROPs (grafting-from). These methods are lengthy in steps, and barely benefit from modern accelerated polymerization strategies. The MOF-based system in our manuscript is a 2-in-1 versatile one: it provides direct and fast route to both pure polypeptides with catalyst recyclability, and straightforward route to polypeptide-MOF hybrid materials with good processability. Importantly, we show that the hybrid material prepared in this way has apparently improved performance than a mixture of the corresponding individual components (MOF and polypeptide). In this revision, we have added new results showing the advantages brought by the strategy in preparing high-performance polypeptide-based hybrid materials.

2. It is not surprising the NCA monomers will decompose when polymerization is carried out over 8h in an aqueous condition in this study. The decomposition products can also initiate NCA polymerization and result in a complex and uncontrollable result. Some detailed studies are expected on these issues.

We thank the reviewer for the comment, but we do not quite get what the reviewer wanted to see here. As we have shown in the manuscript (Figure 4d and 4e), the polymerization in a water-DCM diphasic condition could reach 90% conversion in 4 h, and the polymerization process was highly controllable (Figure S14, unimodal and symmetric peaks), similar to the polypeptides obtained from polymerizations conducted in anhydrous DCM, although the molecular weight could be lower at the same NCA/MOF ratio because of the additional water introduced as initiators. Therefore, we do not think that there is any “complex and uncontrollable result”.

However, the reviewer is correct on the decomposition of NCA in water. The decomposition does not lead to trouble, though. Similar to the “SIMPLE” polymerization reported by Cheng et al, polymerizations initiated by free water or decomposed NCA are not taking place on the MOF surface, and are thus not accelerated. Therefore, these chains hardly propagate and the resulting oligopeptides can be easily removed in the precipitation and washing stage after polymerization. Plus, the water phase actually helps in removing impurities (such as HCl and phosgene) in NCAs, as well as the hydrolyzed NCA (zwitterionic amino acid), therefore it actually reduces the chance of giving “complex and uncontrollable result” in some scenarios.

3. As reported in this study the racemic NCA monomer (D,L-configuration) undergoes polymerization without the self-accelerating cooperative effect, and the monomer conversion is only 68%. For such an incomplete polymerization, how to control the ratio of different subunits in the obtained polymer?

As the catalysis in the chain propagation stage is mainly based on self-acceleration caused by helix array, the system is not suitable for the accelerated polymerization of racemic monomers. When racemic monomers are used, the polymerization rate will be slow. In this study, the racemic monomers are simply listed as a control study to show the necessity of helical structure formation for self-acceleration.

However, the system still allows the preparation of polymers from DL-monomer. When the reaction time is increased (12 h), the monomer conversion can still reach $\geq 90\%$, and the dispersity of the

product is well controlled. Yet, without the acceleration effect, such polymerization will be more sensitive to water.

4. In this study, only ϵ -carboxybenzyl-L-lysine (ZLL) NCA and γ -benzyl-glutamate NCA in L and/or D configurations are used as the monomer for polymerization. It is important to demonstrate that this polymerization method is suitable to other common NCA monomers by incorporating other types of monomers for polymerization.

We thank the reviewer for this advice. We have synthesized another NCA monomer, benzyl-L-aspartate (BLA) NCA, and the polymerizations of BLA-NCA by MOF nanoparticles were also demonstrated to be controllable. Please see the revised Table 1 for more details.

5. To demonstrate the usefulness, it is necessary to study the copolymerization and block polymerization of this polymerization method.

We appreciate this suggestion from the reviewer. To address this issue, we have added the copolymerization and block polymerization data of BLA and BLG in the revised Table 1. The dispersity and isolated yields were presented to demonstrate the wide applicability of this polymerization method.

6. The authors propose that MOF-catalyzed NCA polymerization provides a direct route for the preparation of MOF-polypeptide hybrid materials. However, the advantages of this modified MOF and its potential applications in materials science are not explored or conceptually demonstrated in this study.

We agree with the reviewer. To show the practical utility of this method for preparation of MOF-polypeptide hybrid materials, and the unique advantages that this strategy may offer, we have fabricated the MOF-polypeptide mixed-matrix membrane by this in situ polymerization method. The absorption performance of the prepared membranes was evaluated and compared to membranes prepared from polypeptide, MOF, or their simple mixture. Please see the last section of the revised manuscript for more details.

7. Regarding the catalyst stability, whether this MOF surface modification exists stably?

As MOFs are crystals in nature and their surfaces are actually “defects” in the crystal lattice, there will be uncoordinated metal centers on the surface. These sites require small molecules such as water to “complete” their coordination sphere, therefore water can exist stably on the surface before the addition of reactive NCAs.

The catalyst stability can be experimentally verified in the catalyst recycling experiment. We could recycle the MOF nanoparticles after polymerization by centrifugation and washing, and such recycling could be done for several times. The recycled MOF nanoparticles showed unchanged morphology, and still exhibited good performances on the polymerizations, yielding polypeptides with low dispersity and controllable molecular weights (Figure S6 and revised Figure S17). Thus, it was believed that the catalytic activity of MOF nanoparticles was relatively stable.

8. According to the description in this study, for this polymerization method MOF acts not only as a catalyst but also as a co-initiator to control the molecular weight of the polymerization product.

More demonstration on the control of variable chain length (both shorter and longer chain length) should be studied in Table 1.

We thank the reviewer for this comment. The control of molecular weight by the MOF nanoparticle-mediated NCA-ROPs has been discussed in the manuscript (Figure 6a, 6b), and our results demonstrated that the polypeptides with different molecular weights could be obtained by adjusting the mass ratio of NCA monomers to the nanoparticles (when no additional water was introduced). Detailed analysis including GPC curves, dispersity and mass ratios for the polypeptides with variable molecular weights, prepared through the catalysis of MOF nanoparticles of other sizes, have also been displayed in the Supporting Information (Figure S15). Also, following the reviewer's suggestion, we have added more results showing the molecular weight control into Table 1 (ranging from 4.4 kDa to 143 kDa).

9. The authors mention other types of autocatalytic macromolecular initiators, so the cost of preparing MOF in this paper should be compared with other initiators cited by in this manuscript.

The cost of synthesized MOF and other initiators were displayed as below:

	MOF NPs (Assuming 3 recycles)	Macroinitiator PNB _n (Ref 11)	"SIMPLE" (Ref 15)	PAMAM G2-G5 (Ref 18)	TBAA (Ref 19)
Cost (USD /g)	3.6	189	51.9	547-2623	22.2

This table has been added to the Supporting Information as Table S3.

10. The authors propose that the MOF system can be cycled, providing polymerization results after three cycles. However, the molecular weight of the obtained polymers after cycling increase significantly each time, indicating the loss of activation sites. Whether the MOF catalyst fails largely after more cycles? How will the recycled MOF catalyst control the molecular weight of the polymers?

We thank the reviewer for this comment.

Indeed, according to the polymerization results of the MOF nanoparticles within three cycles, the decreasing molecular weights of the synthesized polypeptides indicated the loss of active sites on the MOF nanoparticles' surfaces during each recycling process. This is almost inevitable for a polymerization catalyst, because polymers can "stick" to the catalyst surface more easily by showing multivalent interactions with the catalyst surface structures. In our opinion, the capability of mediating highly controlled NCA-ROPs after 3-5 cycles already indicates excellent performance. After 5 cycles the catalyst became impractical for use, presumably because of limited remaining catalytic sites. However, according to SEM analysis of such recycled catalyst, their morphology remained identical, indicating the MOF scaffold was very stable in the polymerization and recycling process. Therefore, there is likely room for further improvements in the catalyst regeneration part, restoring the activity of the surface metal sites with appropriate conditions.

To answer the second question from the reviewer, we have provided additional data showing the performance of the recycled MOF nanoparticles on the control of molecular weights for the synthesized polypeptides (Figure S18 in the Supporting Information). The results showed that except for the slightly increased molecular weights, the recycled MOF nanoparticles could still excellent control over molecular weights of the polypeptides in the same way as the original MOF nanoparticles could. With that, the molecular weight increase could be easily countered by slightly decreasing NCA/MOF ratio in the polymerization in the practical use of such recycled MOFs.

Figure S18. The well-controlled polymerization of BLG-NCA by the recycled UiO-66 nanoparticles with varied mass ratio of BLG NCA to the UiO-66 nanoparticles. $[\text{BLG-NCA}]_0 = 0.114$ M, anhydrous DCM as solvent. (a) The GPC-dRI curves of the obtained polypeptides. (b) Comparison on M_n of the obtained PBLGs.

11. Because the MOF catalyst and the polypeptide can be separated by centrifugation and acidification, it is a concern that during polymerization the polypeptides be partially detached from the MOF catalyst and result in both defect in the catalyst surface and soluble polypeptide in the solution. It is likely the defect on the catalyst surface can diminish the acceleration effect in polymerization. The dissolved polypeptide in solution will continue the polymerization process but with different speed compared to that on the MOF surface. How will these affect the controllability of this polymerization method?

We understand the concerns from the reviewer, and the concern is certainly a justified one. However, in practice, we did not see deteriorating controllability of the polymerization resulted from chain detachment. In fact, the detachment of polypeptides from the MOF nanoparticles' surfaces were not that easy: it requires over 13500 RCF centrifugation for more than 30 min, or the even harsher HF treatment (the whole MOF core will be dissolved, forming soluble fluoride salt). As one can see from our data, the polymerizations were always controllable and reproducible, therefore we do not consider chain detachment a real issue that requires serious attention in our system.

12. As a related question, what is the percentage of polypeptide detachment from the surface of catalyst during polymerization? The polymer fraction in Fig 2a after centrifugation shows some turbidity, which indicates possible residual catalyst component in the polymer product.

As we have stated above, we believed that the amount of polypeptides detached during the polymerization was negligible.

The “turbidity” in Figure 2a was actually caused by the lighting effect in the plastic tube, as the plastic tube is not 100% transparent like glassware is. With this revision we have replaced the images in Figure 2 with newly taken photos, and they look more transparent. A photo of BLG-NCA solution in plastic centrifuge tube is also provided, and one can see what a transparent solution looks like in the plastic tube.

Figure 2a. Photos taken during the course of polypeptide preparation by UiO-66 NP-initiated NCA polymerization.

13. Other minor issues that need to be explained

- The authors should explain the reason for the shoulder peaks in GPC in Fig 2c.
- All MALDI peaks in Fig 3a and 3c should be assigned.
- In addition to MALDI, the polymerization kinetics of ethanol-initiated and water-initiated should be compared.

We thank the reviewer for pointing out these issues. Our responses are below:

a) The small shoulder peaks appearing in some of the curves presented throughout the article and the Supporting Information were probably due to the unsynchronous initiation of the NCA-ROP on the MOF nanoparticles' surfaces. This was likely resulted from the heterogeneous nature of this initiation system, and may be related to stirring speed, concentration and other factors. As one can see from the data, the shoulder peak was not always present. This also suggested possible optimization (such as stirring rates, reaction volume, etc.) in the polymerization procedure for the removal of this should peak issue.

We also want to note that the dispersity calculations had taken these shoulder peaks into account. The shoulders were small and exerted very limited impact on the overall dispersity. As the dispersity values were all low, it is still a well-controlled ROP system.

b) We have assigned all the major MALDI peaks in Figure 3a and 3c (now as Figure 4a, 4b). In the figure, formulas were given to indicate the m/z for the assigned peaks (labeled with DP values on the top). In the newly added Figure S7 and S8, more detailed assignments were given.

c) It was unfortunate that we were not able to individually monitor the kinetic of ethanol-initiated polymerization, since it is impossible to achieve 100% replacement of water with ethanol on the

MOF surface. Therefore, we could not eliminate the effects of water in the ethanol-initiated polymerization, and an accurate kinetic study is impossible. Please note that the ethanol-initiated polymerization is a supportive experiment to show the mechanism of the NCA-ROP initiation, and it is not actually used in polypeptide preparation.

Reviewers' Comments:

Reviewer #1:

Remarks to the Author:

The authors have adequately addressed my concern and I am happy for the manuscript to be accepted by Nature Comm.

Reviewer #2:

Remarks to the Author:

This revised manuscript basically addressed all my previous concerns. All the questions have been brightly addressed in this revision. As ultimate minor correction:

In the following sentence "... with Lewis acidity are carbonyl activators, and can serve as catalysts for the ring-opening polymerization of cyclic esters, carbonate and O-carboxyanhydrides through a coordination-insertion mechanism...", the authors should take into account previous work related to Lithium.

I then recommend the manuscript for publication in Nature Communications and I greatly congrats all the authors for their contribution to the field and for their hard work.

Reviewer #3:

Remarks to the Author:

In this revision, the authors answered most of the questions the reviewer raised before. However, some of these answers still didn't address my major concerns on this water-initiated NCA polymerization catalyzed by MOF nanoparticles. The work in this study represents augmentation of previous studies on NCA polymerization, therefore, is lack of novelty for publication in Nature Communications.

Some main issues are as follows.

1. There are many shortcomings in polymerization methods. First, the MOF-catalyzed NCA polymerization method is not universal, although the self-accelerating effect of helical polypeptides in chain propagation stage was used in combination with MOF catalysis in this method. This method cannot achieve various requirements for the structure design of polypeptides, such as functional end group modification. Second, the activation sites on the MOF surface are unknown, therefore the DP of the prepared polypeptides are always unpredictable. Only by exploring the relationship between the NCA/MOF ratio and the DP of the synthesized polypeptides in advance can the synthesis of polymers with different chain lengths be designed on purpose, which is quite troublesome and does not meet the advantages of high controllability. Under the same NCA/MOF ratio, whether the MOF catalysts synthesized in different batches can maintain the same DP of the synthesized polypeptides? These defects about this polymerization strategy are unlikely to be solved with further optimization. The finding in this study doesn't meet the high standard for publication in Nature Communications and should be published in a more specialized journal.

2. In terms of the preparation of polypeptide-MOF hybrid materials, this strategy still has major drawbacks. First, polypeptides are mainly combined with MOF through non-covalent bonds, such as hydrogen bonding. The binding force is very weak, which will greatly affect the stability of the hybrid materials and limit its application. For example, functional studies of polypeptide-MOF hybrid materials

sometimes require the removal of protective groups on the side chains. The deprotection operation is likely to lead to the detachment of MOF and polypeptides, affecting the expected performance of hybrid materials. Second, the authors claimed that the facile preparation of hybrid materials by MOF-catalyzed NCA polymerization is one of innovations, but the adsorption capacity of the hybrid materials shown in Figure 7 is not enough to reflect the important study value of the hybrid materials. The authors should provide more functional applications of this kind of hybrid materials.

There are other issues that the authors have answered previously. But they did not completely solve it.

(a) For the previous question 10, the PBLG solution after centrifugation still shows a little turbidity in figure 3a. I suggest using Dynamic Light Scattering (DLS) to exclude the possibility of MOF nanoparticles residues in the supernatant.

(b) The question 12 answered by the authors before didn't solve my doubts. The authors believe that the detachment of polypeptides from the MOF surface during the polymerization reaction is negligible, which requires more verification. For example, the polymerization mixture needs to be filtered to detect the content of PBLG in the filtrate.

(c) For the previous question 13, the authors didn't explain the reason convincingly for the shoulder peaks in GPC in figure 3 and need to explore an optimized polymerization procedure.

Response to Reviewers

Reviewer #1:

The authors have adequately addressed my concern and I am happy for the manuscript to be accepted by Nature Comm.

We thank the reviewer for all his/her input during the manuscript's revision process.

Reviewer #2:

This revised manuscript basically addressed all my previous concerns. All the questions have been brightly addressed in this revision. As ultimate minor correction:

In the following sentence "... with Lewis acidity are carbonyl activators, and can serve as catalysts for the ring-opening polymerization of cyclic esters, carbonate and O-carboxyanhydrides through a coordination-insertion mechanism...", the authors should take into account previous work related to Lithium.

I then recommend the manuscript for publication in Nature Communications and I greatly congrats all the authors for their contribution to the field and for their hard work.

We thank the reviewer for this comment. Additional references related to lithium catalysis are added based on this suggestion (refs 49, 50, 54). We thank the reviewer for all his/her input during the manuscript's revision process.

Reviewer #3:

In this revision, the authors answered most of the questions the reviewer raised before. However, some of these answers still didn't address my major concerns on this water-initiated NCA polymerization catalyzed by MOF nanoparticles. The work in this study

represents augmentation of previous studies on NCA polymerization, therefore, is lack of novelty for publication in Nature Communications.

We thank the reviewer for these critical comments. In this round of revision, we have performed numerous new experiments to show the versatility of this polymerization approach, based on the reviewer's comments. We hope that these interesting results can address the reviewer's concerns.

Some main issues are as follows.

1. There are many shortcomings in polymerization methods. First, the MOF-catalyzed NCA polymerization method is not universal, although the self-accelerating effect of helical polypeptides in chain propagation stage was used in combination with MOF catalysis in this method. This method cannot achieve various requirements for the structure design of polypeptides, such as functional end group modification.

In the previous round of revision, we have provided additional data showing the polymerization results using different NCA monomers and their combinations. We have also shown previously that the NCAs can be polymerized with other MOFs. With these data, we believe that the polymerization method is robust and universal enough.

To further show the universality of this strategy, and to address the concerns on the application side, we performed a ZIF-8-based polymerization to prepare a degradable mixed-matrix membrane material for sustained release of biocides. Details of this part are provided in the later section of this response letter.

We have also performed additional experiments to show the possibility of achieving functional end-group modification. By using nitrophenyl pyrenemethyl carbonate after the polymerization, *N*-terminal modification of pyrene could be facilely achieved like other traditional NCA polymerization methods. Figures and descriptions related to this experiment are now added to the manuscript and Supporting Information (Figure S19).

Figure S19. Left: a schematic illustration of the polypeptides' end-group modification process using the MOF-catalyzed NCA polymerization strategy. Right: a GPC curve overlay verifying the successful end-group modification of PBLG from the MOF-catalyzed NCA polymerization, using the amine-reactive pyrenemethyl nitrophenyl carbonate.

Second, the activation sites on the MOF surface are unknown, therefore the DP of the prepared polypeptides are always unpredictable. Only by exploring the relationship between the NCA/MOF ratio and the DP of the synthesized polypeptides in advance can the synthesis of polymers with different chain lengths be designed on purpose, which is quite troublesome and does not meet the advantages of high controllability. Under the same NCA/MOF ratio, whether the MOF catalysts synthesized in different batches can maintain the same DP of the synthesized polypeptides? These defects about this polymerization strategy are unlikely to be solved with further optimization. The finding in this study doesn't meet the high standard for publication in Nature Communications and should be published in a more specialized journal.

We understand the concerns by the reviewer regarding to the batch variation and DP control. For the question on the MOF's batch effect on DP variation, the answer is yes, the MOF catalysts synthesized in different batches can maintain almost the same DP of the synthesized polypeptides from a specific NCA monomer. The facileness and reproducibility of MOF preparation is well known. An experiment is now added to the

Supporting Information as Figure S18 to show the batch-to-batch reproducibility. Such consistency is very important for the reproducibility of the synthetic protocol, and for the controllability of the synthesis.

Figure S18. The molecular weights of PBLG synthesized by different batches of UiO-66 nanoparticles using different BLG-NCA/MOF mass ratios. For all the products, the measured dispersity was < 1.1.

In addition, it should be noted that for all the accelerated NCA polymerizations, including ROPISA, SIMPLE and others, the initiation efficiencies are relatively low and the final DP cannot be predicted simply from the M/I ratios used. However, if a polymerization technique can maintain satisfactory reproducibility, the controllability is guaranteed. Such reproducibility is achieved by all the mentioned acceleration polymerization techniques, and by our strategy as well: polymers of different lengths can be obtained at will. We believe that this is what a practical synthetic method needs. In fact, adjusting M/I ratio is not the only common way to control DP in polymerizations (e.g., limiting monomer conversion in radical polymerizations). Controllability can even be achieved by many polymerizations far away from being living (like some hyperbranched polymerizations), as long as the polymerization results are highly reproducible under different conditions. Controllability should not be interpreted simply as DP predictability from M/I ratio, especially in larger-scale polymerizations.

From all the above-mentioned results, the potential batch variation of the MOF-catalyzed NCA polymerization system is considerably low. The UiO-66-initiated NCA polymerization is highly controllable to generate homogenous polypeptides with narrow DP. Considering the preparation of UiO-66 is facile, well-established, and highly

reproducible, we believe that the controllability of our system is higher than the SIMPLE or ROPISA systems, meanwhile being more versatile and far more economical. The following sentence has been added to the manuscript:

“As the facileness and reproducibility of MOF preparation is well known, batch variations for MOF NPs are small enough to allow all the heterogeneously catalyzed polymerizations to be performed with satisfactory consistency and controllability: the MOF catalysts synthesized in different batches can maintain almost the same DP of the synthesized PBLG from BLG-NCA (Figure S18).”

2. In terms of the preparation of polypeptide-MOF hybrid materials, this strategy still has major drawbacks. First, polypeptides are mainly combined with MOF through non-covalent bonds, such as hydrogen bonding. The binding force is very weak, which will greatly affect the stability of the hybrid materials and limit its application. For example, functional studies of polypeptide-MOF hybrid materials sometimes require the removal of protective groups on the side chains. The deprotection operation is likely to lead to the detachment of MOF and polypeptides, affecting the expected performance of hybrid materials.

As we have proposed in our mechanism, the polypeptides are linked to the UiO-66 MOF through oxygen-metal (Zr-O) bonds. By definition, these are coordination (dative) bonds but not hydrogen bonding. To demonstrate the formed mixed matrix membrane (MMM) materials are robust and functional, we have performed more experiments (see our response to the next question). As shown in these results, the strategy is reliable to support the applications of formed MOF-polypeptide as MMM materials without compromising the performance. The examples also included a case involving Zn-O bonds (using ZIF-8 as initiating MOF). In the revised manuscript, we have made a change to indicate the nature of the MOF-polypeptide linkage:

“...the coordinated water molecules in close proximity can attack the carbonyl to give orthoesters which eventually eliminate to give the ring-opened products linked to the MOF surface through Zr-O dative bonds.”

In addition, even if components in a hybrid material are linked through non-covalent interactions, it does not mean that its applications are hindered. Taking MOF-polymer

hybrid material as an example, the traditional method for preparing MOF-polymer MMM is “mixing and casting”, without generating any strong covalent bonds between the MOFs and the polymers. There have been many interesting and meaningful applications derived from such a method (*Chem. Rev.* **2020**, *120*, 8267). Indeed, sometimes the non-covalent compositing can lead to certain issues, and our approach here has provided a novel and straightforward way for generating covalently bonded MMMs. There are already results provided in the manuscript to show the advantage of such covalent strategy.

Finally, we do not consider the deprotection of polypeptide side-chains a relevant problem, since the harsh conditions of many protecting groups utilized in NCA chemistry, such as Cbz and benzyl groups, are well-known and almost irreplaceable (because NCAs are made in an acidic environment). Not just the MOF-polymer covalent linkages, but the MOFs themselves, can be unstable in such deprotecting conditions. For example, ZIF-8 is acid-sensitive. Resolving the deprotection issues requires advances in organic chemistry, and is beyond the scope of this report. There have already exciting strategies reported as solutions to such deprotection issues, such as Lu’s report on a strategy for the synthesis of unprotected NCAs, such as glutamic acid NCA and cysteine NCA without side-chain protections (*Nat. Commun.* **2021**, *12*, 5810). Works like these are probably the viable path toward solutions to the polypeptide deprotection issues. Besides, others (as seen in the above-mentioned review, and many recent reports) and us (as seen in the new section added to the manuscript) have shown that polymers without active functional groups (e.g., amines and carboxylic acids, etc.), such as polyesters, polyolefins, polyacrylates and those not-protected polypeptides, can themselves be combined with MOFs to yield hybrid materials with interesting features.

Second, the authors claimed that the facile preparation of hybrid materials by MOF-catalyzed NCA polymerization is one of innovations, but the adsorption capacity of the hybrid materials shown in Figure 7 is not enough to reflect the important study value of the hybrid materials. The authors should provide more functional applications of this kind of hybrid materials.

We thank the reviewer for this comment. We have added a new section and a figure (Figure 8) to the manuscript to show other potential usages and advantages of hybrid materials

(used for preparation of MMMs) made from this strategy, including the utilization of two new MOFs as co-initiating systems. Examples include the following:

(1) UiO-66@PBLG can serve as a catalyst, aiding the hydrolysis of organic phosphates again utilizing the MOF's Zr centers. This has been demonstrated with a coumarin-based fluorogenic reaction.

(2) Additional metal catalytic centers (in addition to Zr) can be introduced into the MOF strategy to enable other catalytic capabilities. We prepared a copper-containing UiO-67 MOF with 2,2'-bipyridine as the secondary coordination sites for the binding of copper ions. In this UiO-67-Cu MOF, Zr_6 clusters serve as the initiate sites for NCA polymerization while Cu sites can catalyze CuAAC reactions. The UiO-67-Cu was used to polymerize BLG-NCA to obtain UiO-67-Cu@PBLG, and we demonstrated that this MMM material could feature active Cu sites and catalyze a fluorogenic CuAAC reaction.

(3) MOFs can be used as storage materials for the sustained release of molecules of interest. We prepared ZIF-8 with encapsulated 2-methyl-4-chlorophenoxyacetic acid (MCPA, a powerful, selective, widely used herbicide), and this MCPA-containing ZIF-8 could be used to prepare the corresponding MMM. The final MMM showed astonishingly better capability for herbicide sustained release (72% release in 30 days), while the MCPA-containing MOF showed a burst-release profile (95% release in 2 days).

With these new examples, we hope we have demonstrated the versatility of this polymerization methodology derived from MOFs. There are more possibilities though. A review on MOF-polymer hybrid materials (*Chem. Rev.* **2020**, *120*, 8267) has presented an excellent summary of this topic, and this paper is now added as a reference (Ref 62) to help the readers get to know the status of this field.

There are other issues that the authors have answered previously. But they did not completely solve it.

(a) For the previous question 10, the PBLG solution after centrifugation still shows a little turbidity in figure 3a. I suggest using Dynamic Light Scattering (DLS) to exclude the possibility of MOF nanoparticles residues in the supernatant.

We have stated previously that the "turbidity" was actually a lighting effect caused by the plastic centrifuge tube. This time we photographed the solution after it was transferred to

a glass vial. In this way, it is quite clear that the solution is not turbid. Consistently, DLS could not detect MOF particles in the solution. In this revised manuscript, we have again updated Figure 3 with new photos to make this point clear.

(b) The question 12 answered by the authors before didn't solve my doubts. The authors believe that the detachment of polypeptides from the MOF surface during the polymerization reaction is negligible, which requires more verification. For example, the polymerization mixture needs to be filtered to detect the content of PBLG in the filtrate.

We thank the reviewer for this comment. Filtration was a way we had considered before, but it was neither a valid nor convincing way to answer this question. The MOF nanoparticles we typically use are very small (less than 200 nm in diameter), therefore most filtration membranes cannot retain the particles. To solve this issue, we changed to our largest MOF particle (550-600 nm in diameter) for the polymerization, in this way common syringe filters (0.22 or 0.45 μm ones) can be used for filtration. However, it should be noted that the filtration process itself (requiring decent pressure) applies a strong shear force to the structures on nanoparticles, and similar to centrifugation, such force can lead to the detachment of the polypeptides. This process is like the detachment of conjugated proteins on gold nanoparticles during filtration. Therefore, even if polypeptides are detected in the filtrate, it does not necessarily mean that there have been free polypeptide chains during polymerization.

Nonetheless, we have performed the filtration experiment. By forcefully passing the MOF@PBLG mixture through a 0.22 μm PTFE filter, approximately only 20% polypeptide was found detached from the MOF. Considering the sensitivity of this MOF@PBLG material to mechanical force (centrifugation), and the excellent polymerization control with an accelerated polymerization mechanism, we tentatively

believe that the polymer chains mostly remain attached to the MOF surface during the polymerization process.

This part has been added to the Supporting Information as a discussion part. We thank the reviewer for his/her input on this issue.

(c) For the previous question 13, the authors didn't explain the reason convincingly for the shoulder peaks in GPC in figure 3 and need to explore an optimized polymerization procedure.

We sincerely apologize for not being able to explain it clearly. As previously mentioned, the presence of shoulder peaks appearing sometimes in the GPC-dRI curves was due to the unsynchronous initiation of the NCA-ROP on the MOF nanoparticles' surfaces. This was likely resulted from the heterogeneous nature of this initiation system, and may be related to stirring speed, concentration, and other factors. As requested by the reviewer, in this round of revision, we tried to dig more into the reason and attempted to have a better polymerization procedure.

We found that the shoulder peaks were more likely to appear when the stirring was not consistent. For example, when several reaction vials were bundled together with a rubber band and placed on a stirring plate together for the polymerization, shoulder peaks were likely to appear. The stirring bars in different vials might have affected each other, leading to inconsistent stirring that reduces reaction mixture homogeneity, thus the emergence of the shoulder peaks. To verify if this was the issue, we tried to separate the vials in the polymerization stage so that the stir bars would not affect each other's rotation (stirring rate = 800), and the result is shown in the figure below (left). This simple change to the protocol resulted in completely disappeared shoulder peaks in the GPC-dRI curves for polymerizations at lower NCA/MOF ratios, and significantly reduced the shoulder peak intensity at high NCA/MOF ratios. Clearly, the remaining shoulder peaks were caused by catalyst heterogeneity that was not resolved by stirring. The reason we speculated was the partial aggregation of the MOF particles. When NCA/MOF ratio was high, NCAs competed more for the reactive sites on the MOF, thus the effect of aggregation was magnified.

To address this issue, we further tried to pre-disperse the MOF NPs in the polymerization solvent by applying ultrasonic treatment for 40 minutes before monomer addition. This pre-dispersion technique again improved the shape of the GPC peaks, leading to diminished shoulder peaks. Therefore, our findings suggested that the polymerization should be conducted with consistent stirring. In addition, disassembling the aggregated MOF NPs through ultrasonic dispersion could be very useful for better control over the polymerization process when NCA/MOF ratios were high.

Once again, we express our gratitude to the reviewer for his/her valuable feedback, which has allowed us to refine our polymerization protocol and data presentation. We have now included the above relevant data and explanations in the revised manuscript and Supporting Information, and updated our polymerization protocol.

Reviewers' Comments:

Reviewer #3:

Remarks to the Author:

This revised manuscript basically addressed all my previous concerns. I still have some suggestions. The author claimed that metal-organic framework (MOF) nanoparticles provide a heterogeneous catalytic system for the ring-opening polymerization of α -amino acid N-carboxyanhydrides (NCAs). However, MOFs play a catalytic role only in the initiation stage. Since MOF equivalents regulate the molecular weight of the polymer, MOFs seem not to be a catalyst, but more like a co-initiator. Therefore, the description of "MOF-based heterogeneous catalyst" is not accurate. I suggest the author should change this description.

There are also some format errors, such as "Mn" in Table 1. The format of references is inconsistent. For example, in reference 2, all the authors are listed, but the authors are omitted in reference 3, please unify the reference format.

Response to Reviewers

Reviewer #3:

This revised manuscript basically addressed all my previous concerns. I still have some suggestions. The author claimed that metal-organic framework (MOF) nanoparticles provide a heterogeneous catalytic system for the ring-opening polymerization of α -amino acid N-carboxyanhydrides (NCAs). However, MOFs play a catalytic role only in the initiation stage. Since MOF equivalents regulate the molecular weight of the polymer, MOFs seem not to be a catalyst, but more like a co-initiator. Therefore, the description of “MOF-based heterogeneous catalyst” is not accurate. I suggest the author should change this description.

This is an intriguing comment, and we agree with the reviewer that the description is not accurate enough.

MOF equivalents regulate the molecular weight of the polymer because the polymerization is initiated by the coordinated water on MOF, and the amount of coordinated water on MOF is related to MOF used if no external water is added. Adding water to the system also regulates the molecular weight of the polymer. MOF is still a catalyst, but like the reviewer has suggested, it only catalyzes the initiation step. It is indeed odd to call it a ROP catalyst if it only catalyzes the very first initiation reaction.

We feel that it is also not quite accurate to call it a co-initiator, because it does perform a catalytic role in one of the steps in the polymerization, and it can be recycled to do that again. Conversely, in a co-initiation system, the co-initiators usually react with each other and get consumed to form the “real initiator”, as exemplified by BPO/*N,N*-dimethylaniline system for radical polymerizations. The MOF in our case is more like a “carrier” for the initiator (water) rather than being part of it, or its precursor. Also, although the accelerated polymerization is catalyzed by the polypeptides themselves, the MOF particles play an important role in it as well, by “holding” the polypeptide helices together to form an array, which is necessary for the autocatalyzed polymerization. It can still be viewed as a part of the catalytic system.

Therefore, after having weighed our words for some time, we feel that it is most appropriate to call it a “MOF-based heterogeneous catalytic system” instead of a “MOF-based heterogeneous catalyst” (as written in our old manuscript title). The MOF nanoparticles are more suitably called co-catalysts, and are part of the heterogeneous catalytic system. The full catalytic system consists of the MOF particles and the polypeptide helices, and is formed in situ (thus this NCA-ROP is partly autocatalytic).

Based on this thought, we have made the following changes in the manuscript:

Title: changed “Catalyst” to “Catalytic System”

Introduction part: changed “...we show that nanoscale metal-organic frameworks (MOFs) can be utilized as such heterogeneous catalysts. As the first heterogeneous catalyst for NCA-ROP, the nanoscale MOFs, as represented by UiO-66 nanoparticles, feature...” to “...we show that nanoscale metal-organic frameworks (MOFs) can be utilized as heterogeneous co-catalysts. As part of the heterogeneous catalytic system for NCA-ROP, the nanoscale MOFs, together with the in-situ formed polypeptide helix arrays, enable...”

Results and Discussion

Catalyst Design part: changed “...the catalyst can just be responsible for the initiation part, while the chain propagation is aided by the array of the helix formed upon chain initiation...” to “...a co-catalyst can just be responsible for the initiation part, while the chain propagation

is aided by the in-situ formed array of polypeptide helices generated after chain initiation...”

There are also other, similar word changes not listed here (“catalyst” changed to “co-catalyst”, etc.), but all the changes have been highlighted in the manuscript file. Changes are also made to the figures to better indicate the role of MOFs.

We want to thank the reviewer again for this input. Although we eventually decided to use the terms of “catalytic system” and “co-catalyst” instead of “co-initiator”, we do highly value this one comment that had made us to reconsider the term to be used. This has been a very valuable contribution to the refinement of the manuscript.

There are also some format errors, such as “Mn” in Table 1. The format of references is inconsistent. For example, in reference 2, all the authors are listed, but the authors are omitted in reference 3, please unify the reference format.

We checked published *Nat. Commun.* papers and changed “M_n” to “*M_n*” (italicized “M”) throughout the manuscript to make it consistent with journal formatting standard.

The reference format issue is actually from the standard Nature referencing style, used also by *Nat. Commun.* The style rule is: all authors should be included in reference lists unless there are six or more, in which case only the first author should be given, followed by 'et al.'.

Again, we want to thank the reviewer for all his/her time in helping us improve the quality of this manuscript.